# RaBitQCache: Rotated Binary Quantization for KVCache in Long Context LLM Inference

**Wenhao Li** [1]  **Jinhao Dong** [1]  **Hailin Zhang** [2]  **Wenhang Shi** [1]  **Wei Lu** [1]  **Xiaoyong Du** [1]

## Abstract

Long-context Large Language Model inference is severely bottlenecked by the massive Key-Value (KV) cache, yet existing sparse attention methods often suffer from static fixed-budget (Top-k) retrieval or rely on proxy scores that are computationally expensive and biased. To address these limitations, we propose RaBitQ-Cache, a novel sparse attention framework that utilizes randomized rotated binary quantization and high-throughput binary-INT4 arithmetic to efficiently estimate attention weights. Our proxy score serves as an unbiased estimator with a proven error bound, enabling adaptive Top-p retrieval that dynamically adjusts the token budget based on actual attention sparsity. We further implement a hardware-aware system with asynchronous pipelining and lazy updates to mask overhead. Evaluations demonstrate that RaBitQ-Cache significantly accelerates inference and reduces memory I/O while preserving generation quality compared to state-of-the-art baselines. Code is available at `https://github.com/Sakuraaa0/RaBitQCache.git`.

## 1. Introduction

In recent years, Large Language Models (LLMs) have achieved revolutionary breakthroughs in the field of Natural Language Processing (NLP) and, more broadly, in artificial intelligence (Achiam et al., 2023). They are widely utilized in various scenarios, such as video processing, software engineering, and databases (Dosovitskiy, 2020; Fang et al., 2025; Zhou et al., 2024; Hou et al., 2024). As application scenarios become more complex, the maximum context length that LLMs need to support has significantly increased, from the initial 2K-4K to today's 128K or even more (Liu et al., 2025a; Fu et al., 2024; Yang et al., 2025). While the context length continues to grow, the computational complexity of the core component of large models—the self-attention mechanism—has a quadratic relationship with the input sequence length, leading to significant latency overhead (Zhang et al., 2023; Tang et al., 2024). Furthermore, the massive Key-Value (KV) cache considerably increases GPU memory consumption, making it extremely difficult to continuously expand the context length of large models.

Previous studies have shown that the attention calculations in large models exhibit sparsity—only a few key tokens have a decisive impact on the result, while most tokens contribute minimally (Miao et al., 2025; Ge et al., 2023). A natural optimization approach is "selective attention", which involves using only a subset of tokens for computation. Leveraging this characteristic, existing methods are primarily divided into two categories: KVCache eviction (Liu et al., 2023; Zhang et al., 2023) and offloading (Ribar et al., 2024; Zhang et al., 2025; Tang et al., 2024). Eviction methods save storage by discarding early, unimportant key-value pairs, but their assumption that "once unimportant, always unimportant" often does not hold in reality (Dong et al., 2024; Yang et al., 2024a). Offloading methods, on the other hand, retain the data completely and only load a portion of the KVCache for each calculation. In fact, as storage costs continue to decline, storage overhead is no longer a core constraint, making KVCache offloading the mainstream approach (Lin et al., 2025; Wu et al., 2024).

Currently, KV Cache offloading primarily adopts two selection strategies: static and dynamic. The static strategy loads tokens from fixed positions (Xiao et al., 2023b), which lacks adaptivity to input-dependent attention patterns. In contrast, dynamic methods select important tokens on the fly using low-cost proxy scores. However, existing dynamic approaches suffer from two major limitations. **(1) Inaccurate Proxy Scores.** Existing techniques rely on different estimation methods to compute proxy scores, but these methods often fail to reflect true attention importance. For instance, Quest (Tang et al., 2024) selects an entire page at once. Due to the highly discrete distribution of token importance (Xiao et al., 2023a), this leads to the retrieval of numerous irrelevant tokens, causing performance degradation. Similarly,

[1] School of Information, Renmin University of China, Beijing, China [2] Peking University, Beijing, China. Correspondence to: Jinhao Dong <dongjinhao@ruc.edu.cn>.

*Proceedings of the $43^{rd}$ International Conference on Machine Learning*, Seoul, South Korea. PMLR 306, 2026. Copyright 2026 by the author(s).

DS (Yang et al., 2024b) selects partial dimensions for estimation, which is prone to deviating from the true distribution. **(2) Lack of Theoretical Error Bound.** Existing methods provide no theoretical guarantees for their proxy score estimations. This means that the relative values they estimate can only approximate the relative order and cannot accurately estimate the precise values. Consequently, most are limited to fixed token budget Top-$k$ methods and cannot support adaptive Top-$p$ retrieval. However, attention patterns vary significantly across layers and heads—some exhibit highly concentrated distributions, while others are substantially more diffuse (Lin et al., 2025; Xiao et al., 2024; Cai et al., 2024). A rigid Top-$k$ constraint leads to either unnecessary computation or degraded accuracy, making it difficult to strike an optimal balance between efficiency and adaptivity in long-context inference. To achieve Top-$p$ retrieval, proxy scores need to accurately estimate the *magnitude* of attention weights, not merely their relative order.

Our goal is to leverage the sparsity of large language models to achieve inference acceleration by screening key Tokens, while ensuring that model performance is not significantly affected. However, an ideal sparsification scheme must simultaneously satisfy the three core requirements: efficiency, high precision, and generalizability. ❶ The proxy score for evaluating token importance should avoid introducing excessive computational overhead, ❷ the screening mechanism must be precise enough to maintain model performance—ideally possessing a theoretical error bound to guarantee its stability, ❸ and this method needs to be able to adjust quickly and conveniently for different tasks to achieve the best results. Our key insight is that Tokens within the KV Cache are essentially high-dimensional vectors, and the proxy score for assessing their importance is, in fact, a dimensionality reduction operation performed to enable rapid estimation. This perspective naturally leads us to the classic Johnson-Lindenstrauss (JL) Lemma (Johnson et al., 1984). This lemma facilitates the construction of a lightweight proxy score with strict theoretical error bounds, which has found widespread application in Locality-Sensitive Hashing (LSH) and Approximate Nearest Neighbor Search (ANNS) (Gao & Long, 2024; Ailon & Chazelle, 2006; Argerich & Golmar, 2017). Inspired by this, we propose a novel sparse attention framework to resolve existing limitations of existing methods and accelerate inference in long-context scenarios.

In this paper, we propose RaBitQCache, an innovative sparse attention framework. In the prefill phase, we efficiently map Key vectors onto the vertices of a randomly rotated unit hypercube, thereby compressing each high-dimensional Key into a binary vector. In the decoding phase, we construct a novel proxy score that transforms the computationally expensive floating-point vector inner product into an efficient computation between this binary vector and an INT4-quantized Query vector. Crucially, this proxy score serves as an unbiased estimator with a rigorous theoretical error bound, perfectly aligning with the stringent requirement for high precision. This transformation drastically reduces computational overhead, achieving an extremely lightweight design. Ultimately, by leveraging this proxy score for adaptive Top-$p$ retrieval across different tasks, RaBitQCache achieves efficient, precise, and universally applicable acceleration for long-context inference without accessing the full KV cache. Furthermore, we design high-performance kernel operators and system-level scheduling optimizations to maximize hardware utilization and mask latency. Extensive experiments demonstrate that our approach significantly outperforms existing state-of-the-art (SOTA) baselines.

We summarize our contributions as follows:

- We propose RaBitQCache, a novel sparse attention framework with a theoretically-grounded proxy score. It provides an unbiased estimator of attention scores, enabling adaptive Top-$p$ retrieval with a proven error bound.

- We implement efficient kernel operators and system-level scheduling optimizations to maximize hardware utilization and mask latency.

- We conduct a comprehensive evaluation of RaBitQCache, and the results demonstrate that it achieves a $2.16\times$ improvement in efficiency with almost no loss in accuracy.

## 2. Related Work

In this section, we introduce fundamental concepts related to LLMs and Johnson-Lindenstrauss Lemma.

### 2.1. Sparse Attention

The standard attention mechanism computes the output $o$ at step $t$ using the Query $q_t$, Keys $K$, and Values $V$ as $o = \text{softmax}(\frac{q_t K^T}{\sqrt{d}})V$ (Vaswani et al., 2017). Sparse attention approximates this by restricting computation to a subset of indices $I$. Let $M_I$ be a mask matrix where entries corresponding to $I$ are 1 and others are 0. The sparse output is $\hat{o} = \text{softmax}(\frac{q_t K^T}{\sqrt{d}})M_I V$.

To analyze the approximation quality, we bound the error $\|o - \hat{o}\|$. By applying the sub-multiplicative property of the Frobenius norm and leveraging prior findings that the distribution of $V$ is relatively smooth (treating $\|V\|_F$ as a constant) (Zhao et al., 2024), we derive the following bound:

$$
\begin{aligned}
\|o - \hat{o}\| &= \|\text{softmax}\left(\frac{q_t \cdot K^T}{\sqrt{d}}\right)(M_\mathcal{I} - 1^{n \times n})V\| \\
&\leq \|\text{softmax}\left(\frac{q_t \cdot K^T}{\sqrt{d}}\right)(M_\mathcal{I} - 1^{n \times n})\| \cdot \|V\|_F
\end{aligned}
\tag{1}
$$

Consequently, the objective of sparse attention is to select the minimal number of tokens (sparsity) while keeping this error bound as low as possible, thereby mimicking the capabilities of full attention with significantly reduced overhead.

## 2.2. Johnson-Lindenstrauss Lemma

The Johnson-Lindenstrauss (JL) Lemma (Johnson et al., 1984) states that $N$ points in a high-dimensional space can be embedded into a lower-dimensional space of $O(\log N)$ dimensions via random linear projection while approximately preserving pairwise distances. This suggests that computationally expensive high-dimensional retrieval can be optimized through dimensionality reduction, significantly lowering compute costs with minimal accuracy loss.

In this work, we focus on a key corollary of the JL Lemma: in high-dimensional spaces, two random unit vectors are often nearly orthogonal. To ensure the readability of the main text and avoid excessive complex proofs, we only provide a straightforward description here without involving any formulas. The complete derivation of how we leverage this theorem to formulate our unbiased estimator and prove its theoretical bounds is detailed in Appendix A.

## 3. RaBitQCache Execution Process

In this section, we describe the execution process of RaBitQ-Cache. To ensure clarity without delving into the proofs, we first define the operational notations and the core decomposition formula used to approximate the attention scores. Next, we elaborate on the *Prefill* phase for index construction and the *Decode* phase for efficient retrieval. Finally, we elaborate on system-level scheduling optimization strategies. For the detailed derivation of the relevant formulas, please refer to Appendix A.

## 3.1. Notations and Problem Formulation

As established in Section 2.1, identifying the **most significant tokens** is equivalent to finding the key vectors k that yield the largest inner product with a given query vector q. Therefore, our objective is to efficiently estimate the inner product $\langle \mathbf{q}, \mathbf{k} \rangle$ between a query vector $\mathbf{q}$ and key vectors $\mathbf{K}$. Due to the different distributions of query vectors and key vectors in LLMs, we first align them and perform normalization processing. In long-context inference scenarios, the number of tokens in the prefilling phase far exceeds the tokens generated during the decoding phase. Therefore, the vectors produced during the decoding phase do not cause a significant shift in the centroid. Based on this observation, we treat the centroids of the prefilling query and key vectors, denoted as $\mathbf{c}_q$ and $\mathbf{c}_k$, as robust approximations for the global centroids of all data. The normalized vectors are then defined as $\mathbf{q_c} = \frac{\mathbf{q} - \mathbf{c}_q}{||\mathbf{q} - \mathbf{c}_q||}$ and $\mathbf{k_c} = \frac{\mathbf{k} - \mathbf{c}_k}{||\mathbf{k} - \mathbf{c}_k||}$. The original

*Table 1.* Summary of Notations in RaBitQCache

| Symbol | Description |
|---|---|
| $L, D$ | Sequence length and QKV vector dimension |
| $\mathbf{q}_t, \mathbf{K}$ | Query vector at step $t$ and Key matrix |
| $\mathbf{c}_q, \mathbf{c}_k$ | Centroids for Query and Key vectors |
| $\mathbf{q}_c, \mathbf{k}_c$ | Normalized centered Query and Key vectors |
| $\mathbf{P} \in \mathbb{R}^{D \times D}$ | Random orthogonal rotation matrix |
| $\bar{\mathbf{C}}_b \in \{0,1\}^{L \times D}$ | Binary Index (1-bit quantized rotated keys) |
| $\bar{\mathbf{c}}$ | Reconstructed codebook vector from $\bar{\mathbf{C}}_b$ |
| $\mathbf{q}' \in \mathbb{R}^D$ | $\mathbf{q}' = \mathbf{P}^T \mathbf{q_c}$, Query vector after random rotation |
| $\bar{\mathbf{q}} \in \mathbb{R}^D$ | Quantized Query vector (reconstructed) |
| $\bar{\mathbf{q}}_u \in \mathbb{Z}^D$ | Quantized INT4 Query vector (integer) |
| $\alpha \in \mathbb{R}$ | Scalar correction factor for unbiased estimation |

inner product $\langle \mathbf{q}, \mathbf{k} \rangle$ can be decomposed as follows:

$$\langle \mathbf{q}, \mathbf{k} \rangle = \|\mathbf{q} - \mathbf{c}_q\| \cdot \|\mathbf{k} - \mathbf{c}_k\| \cdot \langle \mathbf{q}_c, \mathbf{k}_c \rangle + \langle \mathbf{q}, \mathbf{c}_k \rangle + \langle \mathbf{c}_q, \mathbf{k} \rangle - \langle \mathbf{c}_q, \mathbf{c}_k \rangle \quad (2)$$

In this formula, terms such as $||\mathbf{q} - \mathbf{c}_q||$ and $\langle \mathbf{q}, \mathbf{c}_k \rangle$ are constants computed only once, with their costs amortized over all key vectors. The remaining terms involving only key vectors or centroids, namely $||\mathbf{k} - \mathbf{c}_k||$, $\langle \mathbf{c}_q, \mathbf{k} \rangle$, and $\langle \mathbf{c}_q, \mathbf{c}_k \rangle$, can all be pre-calculated. The computational bottleneck is the normalized dot product $\langle \mathbf{q}_c, \mathbf{k}_c \rangle$. To estimate $\langle \mathbf{q}_c, \mathbf{k}_c \rangle$ with high efficiency and theoretical guarantees. We introduce a random orthogonal rotation matrix $\mathbf{P} \in \mathbb{R}^{D \times D}$. The normalized vectors are rotated and projected onto the vertices of a unit hypercube. Specifically, $\mathbf{k_c}$ is mapped to a binary codebook vector $\bar{\mathbf{c}}$ via the sign of its rotation:

$$\bar{\mathbf{c}} = \frac{1}{\sqrt{D}} \text{sign}(\mathbf{P}^T \mathbf{k_c}) \quad (3)$$

where $\bar{c}$ represents the nearest vertex on the hypercube to the rotated key. Similarly, the query is rotated to $q' = P^T q_c$. Crucially, based on the derivation in Appendix A.3, we construct an estimator for the inner product:

$$\langle \mathbf{q}_c, \mathbf{k}_c \rangle \approx \frac{\langle \bar{\mathbf{c}}, \mathbf{q}' \rangle}{\alpha} \quad (4)$$

Here, $\alpha = \langle \bar{\mathbf{c}}, \mathbf{P}^T \mathbf{k_c} \rangle$ is a scalar correction factor to each key vector. We prove in Theorem A.1 that this formulation serves as an unbiased estimator of the true cosine similarity. Table 1 summarizes the symbols used in this process.

## 3.2. Phase 1: Prefill and Index Construction

The prefilling phase processes the input prompt to generate the initial KV cache and build the search index. Algorithm 1 outlines the construction of the RaBitQCache index.

**Centering and Rotation.** As described in Section 3.1, for an incoming batch of tokens with key vectors $\mathbf{K} \in \mathbb{R}^{L \times D}$,

**Algorithm 1** RaBitQCache Index Construction (Prefill)

1: **Input:** Key tensors $\mathbf{K}$, Rotation Matrix $\mathbf{P}$.
2: **Output:** Binary Index $\bar{\mathbf{C}}_b$, Correction Factors $\boldsymbol{\alpha}$.
3: **Step 1: Centering and Normalization**
4: Calculate centroid $\mathbf{c}_k$ from $\mathbf{K}$
5: **for** each token $i$ in $1 \ldots L$ **do**
6:     $\mathbf{k}^{(i)} \leftarrow \mathbf{K}[i] - \mathbf{c}_k$
7:     $\mathbf{k}_c^{(i)} \leftarrow \mathbf{k}^{(i)}/\|\mathbf{k}^{(i)}\|_2$
8: **end for**
9: **Step 2: Randomized Rotation**
10: $\mathbf{K}' \leftarrow (\mathbf{K} - \mathbf{c}_k)\mathbf{P}^T$
11: **Step 3: Binary Quantization & Correction**
12: **for** each token $i$ in $1 \ldots L$ **do**
13:     $\bar{\mathbf{C}}_b[i] \leftarrow \text{sign}((\mathbf{K}'[i])^T)$
14:     $\bar{\mathbf{c}} \leftarrow (2 \cdot \bar{\mathbf{C}}_b[i] - \mathbf{1}_D)/\sqrt{D}$
15:     $\boldsymbol{\alpha}[i] \leftarrow \langle \bar{\mathbf{c}}, \mathbf{P}^T \mathbf{k}_c^{(i)} \rangle$
16: **end for**
17: Store $\bar{\mathbf{C}}_b$ and $\boldsymbol{\alpha}$

**Algorithm 2** RaBitQCache Decoding with Adaptive Retrieval

1: **Input:** Current Query $\mathbf{q}_t$, Index $\bar{\mathbf{C}}_b$, $\boldsymbol{\alpha}$, Centroids, Matrix $\mathbf{P}$.
2: **Output:** Attention Output $\mathbf{o}_t$.
3: **Step 1: Query Pre-processing & INT4 Quant**
4: $\mathbf{q}_c \leftarrow (\mathbf{q}_t - \mathbf{c}_q)/\|\mathbf{q}_t - \mathbf{c}_q\|_2$
5: $\mathbf{q}' \leftarrow \mathbf{P}^T \mathbf{q}_c$
6: $\bar{\mathbf{q}}_u \leftarrow \text{UniformQuantize}(\mathbf{q}')$
7: **Step 2: Score Estimation**
8: $S_{binary} \leftarrow \text{Int4\_Dot\_Binary}(\bar{\mathbf{C}}_b, \bar{\mathbf{q}}_u)$
9: $S_{norm} \leftarrow \text{Reconstruct}(S_{binary})/\boldsymbol{\alpha}$
10: $S_{orig} \leftarrow \text{Eq.2}(S_{norm})$
11: **Step 3: Adaptive Selection**
12: $I_{select} \leftarrow \text{TopP}(\text{softmax}(S_{orig}/\sqrt{D}), p)$
13: **Step 4: Hybrid Attention**
14: Fetch $\mathbf{K}[I_{select}], \mathbf{V}[I_{select}]$
15: $\mathbf{o}_t \leftarrow \text{Attn}(\mathbf{q}_t, \mathbf{K}_{select} \cup \mathbf{K}_{local}, \mathbf{V}_{select} \cup \mathbf{V}_{local})$

we first calculate the centroid $\mathbf{c}_k$ of the current batch. We then perform centering and normalization to obtain $\mathbf{k}_c^{(i)}$ by subtracting $\mathbf{c}_k$ and normalizing to the unit sphere (lines 6-7). Subsequently, we apply a random orthogonal matrix $\mathbf{P}$, generated once during model initialization, to obtain the rotated keys $\mathbf{K}' = (\mathbf{K} - \mathbf{c}_k)\mathbf{P}$ (line 10). $\mathbf{K}'$ denotes the key vector used for fast codebook computation. Owing to the design of our codebooks, the sign of the selected codebook can be directly determined by the sign bits of $\mathbf{K}'$ (see Eq. 10). Consequently, the key vector can be efficiently represented using binary quantization.

**Binary Index Generation.** We generate the compressed binary index $\bar{\mathbf{C}}_b$ by extracting the sign bit of the rotated vectors, i.e., $\bar{\mathbf{C}}_b = \text{sign}(\mathbf{K}')$ (line 13). This results in a 1-bit representation per dimension. To ensure the estimator remains unbiased, we calculate the scalar correction factor $\boldsymbol{\alpha}[i] = \langle \bar{\mathbf{c}}, \mathbf{P}^T \mathbf{k}_c^{(i)} \rangle$ for each token, where $\bar{\mathbf{c}}$ is the codebook vector derived directly from the binary bits $\bar{\mathbf{C}}_b[i]$ (lines 14-15). Finally, $\bar{\mathbf{C}}_b$ and $\boldsymbol{\alpha}$ are stored in GPU memory to facilitate the rapid subsequent estimation of the inner product using Eq. 4.

### 3.3. Phase 2: Decode with RaBitQCache

The decoding phase generates tokens step-by-step. For each new query $\mathbf{q}_t$, RaBitQCache executes the pipeline shown in Algorithm 2.

**Query Quantization.** From Theorem A.1, we know that the error introduced in the previous Eq. 4 is $O\left(\frac{1}{\sqrt{D}}\right)$. Therefore, to accelerate the inner product computation, we can quantize $q'$ if the error from this quantization step is within the order of magnitude of the above error. According to

Theorem A.2, uniform scalar quantization can be employed to quantize the floating-point vector $q'$ into a 4-bit integer vector $\bar{q}_u$. Subsequently, we can utilize Eq. 17 to accelerate the calculation. Thus, upon receiving $\mathbf{q}_t$, we first center and rotate it to obtain $\mathbf{q}' = \mathbf{P}^T(\mathbf{q}_t - \mathbf{c}_q)$ (lines 4-5). Then, we quantize the floating-point vector $\mathbf{q}'$ into a 4-bit integer vector $\bar{\mathbf{q}}_u$ using uniform scalar quantization (line 6):

$$\bar{\mathbf{q}}_u = \text{Round}\left(\frac{\mathbf{q}' - q_l}{\Delta}\right) \tag{5}$$

where $[q_l, q_r]$ is the dynamic range of $\mathbf{q}'$ and $\Delta = \text{Round}(\frac{q_r - q_l}{2^4 - 1})$ is the step size.

**Hardware-Efficient Score Estimation.** The computationally expensive floating-point vector inner product is transformed into an efficient operation between the binary index $\bar{\mathbf{C}}_b$ and the INT4-quantized query $\bar{\mathbf{q}}_u$. We implement a custom GEMV CUDA kernel to compute the dot product $\langle \bar{\mathbf{C}}_b, \bar{\mathbf{q}}_u \rangle$ (line 8). Then, this result can be used in Eq. 17 to reconstruct the approximate estimated value of $\langle \bar{\mathbf{c}}, \mathbf{q}' \rangle$ (Line 9). For details on how this result is specifically reconstructed, please refer to Appendix A.3.2. By dividing by the correction factor $\alpha$, we obtain the normalized similarity $S_{norm} = \langle \mathbf{q}_c, \mathbf{k}_c \rangle$ (Eq. 4). Finally, substituting this back into Eq. 2 yields the original attention scores $S_{orig}$ (line 10).

**Adaptive Selection and Data Fetching.** Based on the proxy scores, we employ an adaptive sampling strategy. The function $\text{TopP}(\text{softmax}(S_{orig}/\sqrt{D}), p)$ efficiently identifies the minimum set of indices $I_{select}$ required to cover the probability mass $p$ (line 12). The system then fetches the corresponding full-precision KV pairs. Finally, the attention output is computed using a hybrid context of the retrieved sparse tokens and the local sliding window (lines 14-15).

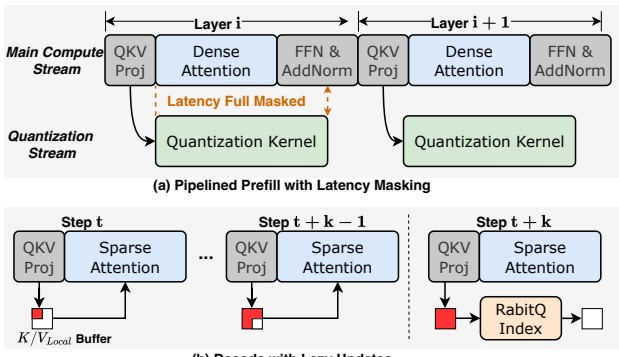

*Figure 1.* Overview of RaBitQCache System-Level Scheduling Optimizations: Asynchronous Pipelined Prefill for Latency Masking and Lazy Updates for Efficient Decoding.

We will introduce how we seamlessly integrate the local sliding window into our system in Section 3.4.

### 3.4. Pipelined Scheduling and Lazy Updates

To further achieve system-level acceleration, we adopt two hardware-aware scheduling strategies for the prefilling and decoding phase based on their computational characteristics. The specific scheme is illustrated in Figure 1.

**Asynchronous Prefill.** We observe that the index constructed during the prefilling phase is only required until the first decoding step of that layer. Leveraging this, we decouple index construction from the main attention computation. While the primary CUDA stream executes the compute-intensive $O(L^2)$ dense attention, a secondary low-priority stream concurrently quantizes the key vectors ($O(L)$) into the binary index. This overlapping ensures the indexing latency is effectively hidden by the quadratic complexity of the prefill attention, resulting in zero visible overhead during prefilling phase.

**Lazy Decode Updates.** In the decoding phase, updating the index token-by-token incurs significant kernel launch overhead. We therefore adopt a Lazy Update strategy where the local window $\mathbf{K}_{local}$ acts as a full-precision write-back buffer. New tokens are temporarily accumulated in this buffer and participate in the hybrid attention calculation $\mathbf{o}_t = \text{Attention}(\mathbf{q}_t, \mathbf{K}_{fetched} \cup \mathbf{K}_{local}, \mathbf{V}_{fetched} \cup \mathbf{V}_{local})$ without immediate quantization. These tokens will only be quantized in batches when $\mathbf{K}_{local}$ reaches the predefined batch capacity. This strategy exploits data parallelism to saturate GPU resources and minimize kernel launch overheads. This optimization not only improves efficiency but also enhances the generation quality. Inherently, this design functions as a sliding window that guarantees high-precision access to the recent context, while the index addresses long-range dependencies via adaptive Top-$p$ retrieval.

---

**Algorithm 3** Top-$p$ Mask Generation Logic (Adapted from FlashInfer)

---

1: **Input:** Scores $S$, Target mass $p$
2: **Output:** Boolean Mask $M$
3: $low \leftarrow 0, high \leftarrow \max(S)$
4: **repeat**
5: $\quad \pi_0 \leftarrow (high + 2 \cdot low)/3$
6: $\quad \pi_1 \leftarrow (2 \cdot high + low)/3$
7: $\quad G_0 \leftarrow \sum_{s \in S} s \cdot \mathbb{I}(s > \pi_0)$
8: $\quad G_1 \leftarrow \sum_{s \in S} s \cdot \mathbb{I}(s > \pi_1)$
9: $\quad v_{min} \leftarrow \min\{s \in S \mid s > low\}$
10: $\quad v_{max} \leftarrow \max\{s \in S \mid s \leq high\}$
11: $\quad$ **if** $G_1 \geq p$ **then**
12: $\quad\quad low \leftarrow \pi_1$
13: $\quad$ **else if** $G_0 \geq p$ **then**
14: $\quad\quad low \leftarrow \pi_0$
15: $\quad\quad high \leftarrow \min(\pi_1, v_{max})$
16: $\quad$ **else**
17: $\quad\quad high \leftarrow \min(\pi_0, v_{max})$
18: $\quad$ **end if**
19: **until** $v_{min} = v_{max}$
20: $\theta \leftarrow low$
21: $M \leftarrow (S > \theta)$

---

## 4. Efficient Kernel Implementations

In this section, we discuss the low-level optimizations employed to maximize the throughput of RaBitQCache.

### 4.1. Efficient Top-$p$ Operator

To ensure high-throughput performance during the adaptive retrieval phase, we implement a custom fused CUDA kernel. Our approach is inspired by the *Nucleus (Top-p) Sampling* strategy commonly used in LLM generation (Ravfogel et al., 2023; Brown et al., 2020). Specifically, we adapt the high-performance sampling kernel from FlashInfer (Ye et al., 2025) to serve as our massive-parallel KVCache retrieval masking operator.

Standard Top-$p$ implementations typically rely on sorting, incurring an $O(L \log L)$ cost that bottlenecks long-context processing. Instead, we employ a linear-time ternary search approach based on value partitioning, as outlined in Algorithm 3. The process begins by initializing a search interval $[low, high]$ based on the min/max values of the score distribution (Line 3). Inside the iterative search loop (Lines 4-19), the algorithm partitions the current value range using two pivots, $\pi_0$ and $\pi_1$, effectively dividing the search space into three segments. For each iteration, we concurrently compute two key statistics: the accumulated probability mass ($G_0, G_1$) above each pivot, and the tightest data-dependent bounds ($v_{min}, v_{max}$) within the current range.

Based on the comparison between the accumulated mass $G$ and the target $p$, we dynamically discard parts of the search space (Lines 11-18). This loop continues until the data-dependent bounds converge ($v_{min} = v_{max}$), ensuring the final threshold $\theta$ precisely matches an existing score in the distribution. Finally, a boolean mask is generated by comparing all scores against this derived threshold. For more details about this kernel, please refer to Appendix B.1.

## 4.2. Hardware-Efficient Approximate Score Computation

The core computational bottleneck in RaBitQCache lies in estimating the similarity scores between the quantized INT4 query vector $\bar{q}_u$ and the massive binary key cache $\bar{C}_b$. To maximize the throughput of the approximate score estimation, we implement a custom CUDA kernel that optimizes the INT4 × Binary GEMV operation through three critical techniques:

**Packed Binary Representation.** We address the significant memory bandwidth waste inherent in storing binary keys as 8-bit integers (`uint8`) by adopting a dense storage format. Specifically, we pack 32 consecutive binary elements along the head dimension into a single 32-bit integer, achieving an $8\times$ reduction in memory bandwidth consumption. To prevent runtime latency, this data packing is performed ahead of time during the quantization phase, avoiding dynamic overhead during inference.

**Shared Memory Tiling.** Observing that the same INT4 quantized query vector $\mathbf{q}_u$ is required for inner product calculations against all keys, we implement a caching strategy to eliminate redundant global memory accesses. Each thread block first cooperatively loads the current token's $\mathbf{q}_u$ into on-chip shared memory; subsequently, all threads within the block read from this low-latency cache while processing distinct key-value pairs. This design effectively reduces global memory traffic for the query vector by a factor equivalent to the block size.

**Vectorized Bit Extraction.** To optimize the inner loop of the dot product computation, we leverage vectorized instructions and bitwise arithmetic. We utilize 128-bit vectorized loads to fetch data efficiently and employ fast bitwise operations to extract individual bits from the packed binary format. When combined with loop unrolling, this design enables the compiler to generate a highly efficient instruction pipeline that maximizes instruction-level parallelism.

## 5. Evaluation

In this section, we perform quantitative experiments to demonstrate that RaBitQCache effectively accelerates long-context inference while preserving generation quality. We first describe our implementation and experimental setup in

*Table 2.* Main results on LongBench averaged over 13 datasets. The results are formatted as Generation Score (Attention Recall). The recall rate is defined as the accumulated attention score of the selected tokens divided by the total score of Full Attention. RaBitQ denotes RaBitQCache.

| Method | Budget | Longchat-7B-v1.5-32k | LLaMA-3.1-8B |
|---|---|---|---|
| Full | -(100%) | 35.78(100%) | 50.58(100%) |
| Oracle | 1024 (11.4%) | 35.84(89.3%) | 50.31(86.9%) |
| RaBitQ | $p = 0.95$ (17.33%) | 36.21(89.8%) | **50.6**(90.7%) |
| Quest | 256 (2.84%) | 30.50(27.3%) | 36.71(53.3%) |
| | 1024 (11.38%) | 35.30(58.1%) | 46.52(82.6%) |
| | 4096 (40.29%) | **36.28**(86.1%) | 49.24(92.9%) |
| | 8192 (64.57%) | 36.11(94.7%) | 50.43(97.0%) |
| DS | 256 (2.86%) | 3.75(10.0%) | 49.06(42.7%) |
| | 1024 (11.42%) | 9.52(20.4%) | 50.28(68.4%) |
| | 4096 (40.33%) | 21.28(47.8%) | 50.40(90.7%) |
| | 8192 (64.60%) | 32.58(67.4%) | 50.47(96.9%) |
| SparQ | Ratio=0.25 (25.00%) | 35.91(93.1%) | 50.15(89.8%) |

Section 5.1. We then present the accuracy results in Section 5.2 and the efficiency performance analysis in Section 5.3. Finally, we perform ablation studies in Section 5.4.

### 5.1. Implementation and Setup

We implemented RaBitQCache as a high-performance serving system built upon vLLM v0.10.2 (Kwon et al., 2023), integrating FlashInfer v0.5.3 (Ye et al., 2025) for optimized attention kernels and LMCache (Liu et al., 2025b) for memory management. The system comprises over 5,000 lines of code, utilizing Python for orchestration, OpenAI Triton, and custom CUDA kernels. All experiments are conducted on servers equipped with NVIDIA Hopper-architecture GPUs.

### 5.2. Accuracy Experiments

**Benchmarks and Models.** To evaluate generation quality and retrieval accuracy, we use three complementary benchmarks: **LongBench** (Bai et al., 2024) for realistic long-context QA, summarization, and code understanding; **RULER** (Hsieh et al., 2024) for synthetic long-context retrieval at scales of 8K to 64K; and **GSM8K** (Cobbe et al., 2021) for mathematical reasoning under the long-output regime. We select three widely-used models spanning a wide capacity range: Longchat-7B-v1.5-32k (Li et al., 2023), LLaMA-3.1-8B-Instruct (Dubey et al., 2024), and the larger LLaMA-3.1-70B-Instruct (Dubey et al., 2024).

**Baselines.** We compare RaBitQCache against several state-of-the-art sparse attention, including Quest (Tang et al., 2024), Double Sparse (DS) (Yang et al., 2024b), SparQ (Ribar et al., 2024), MagicPIG (Chen et al., 2025),

*Table 3.* Average LongBench score on LLaMA-3.1-8B-Instruct against additional sparse attention baselines. RaBitQCache uniquely supports adaptive Top-$p$; all other methods are restricted to fixed-budget Top-$k$ or KV cache compression. Method abbreviations: MPIG=MagicPIG, Pyr=PyramidKV, Snap=SnapKV, PQC=PQCache.

| Method | RaBitQ | MPIG | Pyr | Snap | PQC | KIVI |
|--------|--------|------|-----|------|-----|------|
| Avg. | **50.63** | 49.95 | 45.09 | 44.91 | 50.34 | 50.13 |

*Table 4.* Generalization across model scales and benchmarks. LongBench reports average score across 13 tasks (LLaMA-3.1-70B-Instruct); RULER reports average accuracy across 8K–64K (LLaMA-3.1-8B-Instruct); GSM8K reports accuracy / attention recall (LLaMA-3.1-8B-Instruct).

| Benchmark | Full | RaBitQ | Quest | SparQ |
|-----------|------|--------|-------|-------|
| LongBench (Avg.) | 54.62 | 54.58 | 50.69 | 53.76 |
| RULER (Avg.) | 79.65 | 79.55 | 78.20 | 79.08 |
| GSM8K (Acc.) | 0.81 | 0.77 | 0.70 | 0.77 |
| GSM8K (Recall) | 1.00 | 0.888 | 0.813 | 0.605 |

PyramidKV (Cai et al., 2024), SnapKV (Li et al., 2024), PQCache (Zhang et al., 2025), and KIVI (Liu et al., 2024). Following the baselines, we use full attention in the first two layers to ensure fair comparison. For all methods, we use the recommended configurations from their official repositories: Quest (`chunk_size`=16), SparQ ($r = 16$), DS ($r = 32$, `quant_bit`=2, Query channel), MagicPIG ($K = 10$, $L = 150$, $W = 64$), PyramidKV (`capacity`=512, `window`=64), SnapKV (`capacity`=1024, `window`=64), PQCache (`compress`=0.1, `subvec`=2, `bits`=6), and KIVI (`k/v_bits`=2, `group`=32, `residual`=128). For RaBitQCache, we set the hyperparameter $p = 0.95$ on LongBench and $p = 0.9$ on RULER and GSM8K.

**Results on Longbench.** We selected 13 tasks from the LongBench benchmark that cover all task types. To conduct a comprehensive evaluation, we tested multiple token budgets for every baseline scheme. Additionally, we evaluated an "Oracle" method that selects the optimal Top-$k$ tokens based on the ground-truth attention scores calculated via full attention, serving as an upper bound for retrieval-based approaches. The summarized average results are presented in Table 2. For detailed results across specific tasks, please refer to Table 6 and Table 7.

It can be observed that our method incurs negligible performance loss, consistently performing on par with or even surpassing the full-attention baseline. In contrast, other competitive methods exhibit significant performance gaps when constrained by varying token budgets. On the LongChat model, we achieve 89.85% attention recall using only 17.33% of the budget—efficiency nearly identical to the Oracle (89.30% recall with 11.42% budget). Although Quest-4096 marginally outperforms our approach on the LongChat model, it comes at a significantly higher cost (40.29%) and achieves a lower recall rate (86.05%). Detailed analysis shows that token requirements vary drastically across tasks; fixed token budgets lack the flexibility to adapt to these variations, whereas our adaptive retrieval automatically adjusts the budget without parameter tuning.

To further position RaBitQCache against the broader sparse attention literature, we additionally compare with five state-of-the-art baselines on LLaMA-3.1-8B-Instruct, covering retrieval-based (MagicPIG, SnapKV), eviction-based (Pyra-

midKV), and quantization-based (PQCache, KIVI) methods. As summarized in Table 3, RaBitQCache achieves the highest average score among all baselines. More importantly, MagicPIG, PyramidKV, SnapKV, and PQCache only provide *relative* ranking information and therefore can only support fixed-budget Top-$k$ selection, while KIVI is orthogonal to retrieval as it compresses the KV cache directly; in contrast, RaBitQCache provides an unbiased estimate of attention scores with a provable error bound and is the only method capable of supporting adaptive Top-$p$ retrieval. A natural integration is to first apply our method for adaptive selection and then apply KIVI-style compression to the retained tokens, which we leave as future work. Full per-task results are provided in Table 8 of the appendix. These results indicate that our adaptive retrieval mechanism effectively ensures high recall of critical information, maintaining robust performance in all cases.

**Generalization to Larger Models and Diverse Benchmarks.** To further validate the generality of RaBitQCache across model scales and task types, we evaluate on LLaMA-3.1-70B-Instruct (LongBench) and on LLaMA-3.1-8B-Instruct (RULER for synthetic long-context retrieval at 8K–64K, GSM8K for mathematical reasoning with long outputs). On RULER and GSM8K, DS and Quest use a fixed budget of 4096 (RULER) or 256 (GSM8K) tokens, SparQ uses its default setting, and RaBitQCache uses adaptive Top-$p$ with $p = 0.9$. The summarized results are presented in Table 4; full per-task results are provided in Table 9–11 of the appendix.

Several observations stand out. *First*, on LongBench with LLaMA-3.1-70B, RaBitQCache attains 54.58, nearly closing the gap to full attention while clearly outperforming Quest (50.69) and SparQ (53.76). *Second*, on RULER, our method matches or even surpasses Full at 8K–32K and remains competitive at 64K, while spending only 1076 tokens at 8K and naturally scaling to 3581 tokens at 64K—directly demonstrating the task-adaptive nature of Top-$p$ retrieval. *Third*, on GSM8K, RaBitQCache matches the accuracy of DS and SparQ (0.77) while attaining the highest attention recall (0.888) among all sparse baselines, indicating robustness even when the output length grows large.

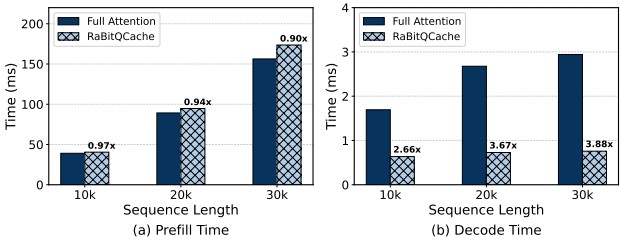

*Figure 2.* Inference Latency Analysis. (a) Prefill: RaBitQCache incurs negligible overhead ($< 10\%$) via asynchronous quantization. (b) Decoding: RaBitQCache achieves up to $3.88\times$ speedup at 30k context, effectively decoupling latency from context growth.

*Figure 3.* End-to-End Latency Comparison. We achieve a maximum $2.16\times$ system-level speedup by significantly reducing memory I/O bottlenecks while maintaining retrieval precision.

### 5.3. Efficiency Experiments

**Benchmarks and Setup.** We evaluate system efficiency by measuring end-to-end latency acceleration and attention computation speedup, specifically distinguishing between the prefill phase (Time-To-First-Token, TTFT) and the decoding phase (Time-Between-Tokens, TBT). We utilize vLLM integrated with FlashInfer, employing FlashAttention-2 (Dao, 2023) as the backend, to serve as our Full Attention baseline. Notably, our preliminary tests indicate that neither the official repositories of other sparse attention systems nor their naive implementations within vLLM can outperform the absolute speed of the highly optimized vLLM-based Full Attention baseline. To address this issue and enable a fair comparison, instead of excluding these baselines, we normalize the execution time of each method against the Full Attention implementation within its respective system. This allows us to evaluate the relative acceleration ratio achievable by each method. The evaluation utilizes real-world workloads from the LongBench dataset from 10k to 30k tokens to assess performance scaling.

**Prefill Latency (TTFT).** Figure 2 (a) compares the Time-to-First-Token. RaBitQCache exhibits negligible overhead (less than $10\%$) compared to the standard FlashAttention baseline. The main performance degradation stems from the fact that Prefill is a computationally intensive task, so asynchronous computing will inevitably affect the efficiency of the main CUDA stream computation. This result empirically confirms the effectiveness of our asynchronous pipelining strategy described in Section 3.4, which successfully masks the computational cost of random rotation and quantization behind the dense attention computation.

**Decoding Latency (TBT).** We evaluate the generation latency across varying context lengths ranging from 10k to 32k. As shown in Figure 2 (b), RaBitQCache achieves a significant speedup of up to $3.88\times$ compared to Full Attention. This consistent acceleration verifies that our lightweight binary index scan effectively alleviates the memory I/O bottleneck inherent in loading massive KVCaches, decoupling

inference latency from context growth.

**End-to-End Speedup.** To assess real-world performance, we also assessed the end-to-end acceleration of different systems. For different methods, we adopt a token budget that achieves comparable scores. As illustrated in Figure 3, RaBitQCache achieves an overall acceleration of $2.16\times$ compared to the full precision baseline. This demonstrates that our method translates theoretical computational reductions into substantial wall-clock speedups, making it a highly practical solution for long-context serving.

### 5.4. Ablation Studies

**Sensitivity to Threshold $p$.** We investigate the sensitivity of the cumulative probability threshold $p$ by evaluating RaBitQCache on the NarrativeQA and HotpotQA datasets. As illustrated in Figure 4, we track the Generation Score alongside the Retrieval Recall (accuracy) and Token Budget (efficiency) as $p$ varies from 0.65 to 0.95. We observe that the response trends to changes in $p$ are highly consistent across different tasks; this stability stems from the fact that $p$ represents the cumulative probability mass, which is less susceptible to task-specific variations. In stark contrast, the static budget $k$ exhibits significant divergence across

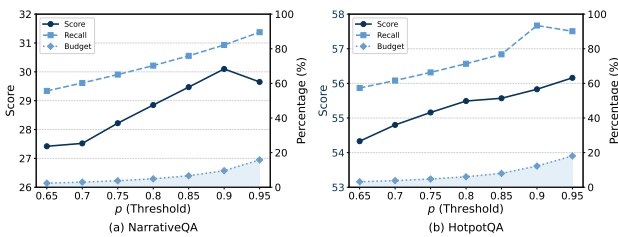

*Figure 4.* Sensitivity analysis of threshold $p$ on (a) NarrativeQA and (b) HotpotQA. The Left Y-axis denotes the generation quality (Score), while the Right Y-axis indicates the percentage for both Recall and Token Budget.

*Table 5.* `Int4DotBinary` operator latency with different configs. The **Config** column corresponds to the tuple (*batch_size*, *num_quantized_tokens*, *num_heads*, *head_size*).

| Config | Original(ms) | Warp(ms) | speedup |
|---|---|---|---|
| 1×1024×8×128 | 0.0382 | 0.0127 | 3.01× |
| 1×4096×8×128 | 0.0386 | 0.0130 | 2.96× |
| 1×8192×8×128 | 0.0385 | 0.0129 | 2.98× |
| 1×16384×8×128 | 0.0388 | 0.0136 | 2.84× |
| 4×4096×8×128 | 0.0387 | 0.0130 | 2.97× |
| 8×4096×8×128 | 0.0741 | 0.0216 | 3.43× |
| 1×4096×32×128 | 0.1585 | 0.0501 | 3.16× |
| 1×8192×8×64 | 0.0212 | 0.0093 | 2.27× |

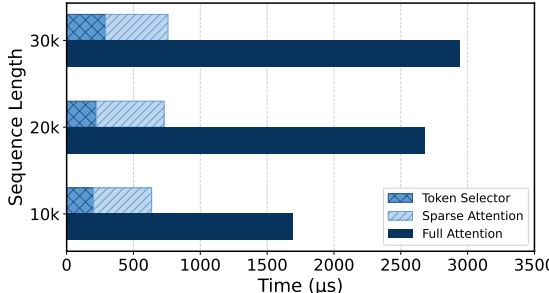

*Figure 5.* Time Breakdown for RabitQCache. The computational overhead of the Token Selector is negligible compared to the savings gained. By filtering irrelevant tokens, RaBitQCache reduces the heavy Attention computation compared to the Full Attention baseline.

tasks, as evidenced in Table 7. Furthermore, RaBitQCache achieves performance comparable to full attention while accessing less than 20% of the KV cache. This empirically validates the effectiveness of our unbiased estimator in accurately identifying sparse attention patterns. These results demonstrate that $p$ serves as a robust, data-dependent hyper-parameter capable of automatically adapting the retrieval budget to the intrinsic information density of the context.

**Effect of Centroid Re-centering.** RaBitQCache uses prefill-phase centroids $c_q, c_k$ to re-center query and key vectors before quantization, which is required by the theoretical derivation that assumes near-uniform distribution on the unit hypersphere. To assess its empirical impact, we compare with a variant that omits the centering step. Across 13 LongBench tasks on LLaMA-3.1-8B-Instruct, removing re-centering reduces the average score from 50.63 to 50.25. Per-task results are reported in Table 12 of the appendix. While the empirical degradation is mild on the tested workloads, re-centering remains theoretically required to satisfy the unit-hypersphere uniformity assumption underlying the $O(1/\sqrt{D})$ error bound. Moreover, recent work (Xing et al., 2026) observes that key and query vectors tend to be tightly clustered rather than uniformly distributed on the unit hypersphere, which may invalidate the uniformity assumption in practice. Whether re-centering is strictly necessary in practical deployment under such clustered distributions and significant decode-phase distribution drift, together with a more rigorous theoretical treatment of these settings, remains an interesting direction for future work.

**Kernel Optimization Efficiency.** To validate the effectiveness of our hardware-aware optimizations, we conduct a micro-benchmark comparing our custom fused `Int4DotBinary` GEMV kernel against a naive CUDA implementation. We vary the Batch Size, Number of tokens, Number of Heads, and Head Size. As shown in Table 5, our optimized kernel delivers consistent acceleration across all configurations, achieving up to 3.43× speedup. These results prove that techniques such as packed binary storage and shared memory tiling effectively maximize memory bandwidth utilization and instruction throughput.

**Time Breakdown for RabitQCache.** To deeply understand the source of our performance gains, we conduct a fine-grained latency decomposition analysis, as illustrated in Figure 5. It is important to note that, as detailed in Section 3.4, the overhead of index construction during the prefill phase is fully masked by the computation-intensive dense attention via our asynchronous pipeline strategy. Consequently, our evaluation focuses exclusively on the latency breakdown during the decoding phase.

Figure 5 presents the time decomposition across different context lengths. From the figure, we can observe that the computational overhead introduced by our algorithm is equivalent to only 10% of the Full Attention baseline. However, RaBitQCache achieves a significant net reduction in total inference latency by drastically reducing the number of tokens required for the heavy full-precision attention computation. This high efficiency is driven by the lightweight nature of our algorithmic design combined with the high-performance implementation of our optimized kernel operators, which ensure that the retrieval overhead remains negligible relative to the computational savings.

## 6. Conclusion

In this paper, we introduced RaBitQCache, a novel sparse attention framework that addresses the memory and computational bottlenecks of long-context LLM inference through randomized rotated binary quantization and high-throughput INT4 arithmetic. By establishing a theoretically grounded unbiased estimator, our method enables precise adaptive Top-$p$ retrieval that dynamically adjusts to varying attention patterns. Extensive evaluations demonstrate that RaBitQ-Cache effectively combines algorithmic innovation with hardware-aware optimizations to achieve significant acceleration and memory reduction compared to state-of-the-art baselines, all while preserving generation quality.

## Acknowledgment

This work is supported by the National Natural Science Foundation of China under Grant 62441230, 62461146205, 62072458 and 62472429. We also sincerely thank the authors of FlashInfer (Ye et al., 2025) and RaBitQ (Gao & Long, 2024), whose prior work has provided inspiration for our research.

## Impact Statement

This paper presents work whose goal is to advance the field of Machine Learning, specifically in optimizing the inference efficiency of Large Language Models (LLMs). The proposed RaBitQCache framework significantly lowers the computational memory and latency barriers for processing long-context information. This contributes to "Green AI" by reducing energy consumption and facilitates the democratization of AI by enabling long-context capabilities on consumer-grade hardware.

However, as RaBitQCache relies on quantization and sparse retrieval approximation, there remains a theoretical possibility of information loss, despite the rigorous error bounds established in our work. In high-stakes applications requiring absolute precision over long documents (e.g., critical medical analysis or legal precedent review), users should remain aware of the trade-off between inference speed and the potential for recall degradation. We believe the societal benefits of efficient inference outweigh these risks, provided that deployment in sensitive domains is accompanied by appropriate validation.

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

# A. RaBitQCache Method Derivation

In this section, we derive the theoretical guarantees for the proxy score we use and prove that the score is an unbiased estimator of the original inner product with a theoretical error bound.

## A.1. Inner Product Decomposition

As established in Section 2.1, identifying the **most significant tokens** is equivalent to finding the key vectors $k$ that yield the largest inner product with a given query vector $q$. To lay the groundwork for subsequent analysis based on the Johnson-Lindenstrauss (JL) lemma, we first normalize all query and key vectors. Let the original query and key vectors be $\mathbf{q}, \mathbf{k} \in \mathbb{R}^D$. In long-context inference scenarios, the number of tokens generated during the prefilling phase vastly exceeds that of the decoding phase. We therefore leverage this property by treating the centroids of the prefilling Q and K vectors, denoted as $\mathbf{c}_q$ and $\mathbf{c}_k$, as robust approximations for the global centroids of all data. The normalized vectors are then defined as $\mathbf{q_c} = \frac{\mathbf{q} - \mathbf{c}_q}{||\mathbf{q} - \mathbf{c}_q||}$ and $\mathbf{k_c} = \frac{\mathbf{k} - \mathbf{c}_k}{||\mathbf{k} - \mathbf{c}_k||}$. This re-centering step ensures a zero-mean data distribution, which is instrumental for the effectiveness of the subsequent quantization process. The original inner product $\langle \mathbf{q}, \mathbf{k} \rangle$ can be decomposed as follows:

$$\langle \mathbf{q}, \mathbf{k} \rangle = ||\mathbf{q} - \mathbf{c}_q|| \cdot ||\mathbf{k} - \mathbf{c}_k|| \cdot \langle \mathbf{q}_c, \mathbf{k}_c \rangle + \langle \mathbf{q}, \mathbf{c}_k \rangle + \langle \mathbf{c}_q, \mathbf{k} \rangle - \langle \mathbf{c}_q, \mathbf{c}_k \rangle \tag{6}$$

For a given query, terms such as $||\mathbf{q} - \mathbf{c}_q||$ and $\langle \mathbf{q}, \mathbf{c}_k \rangle$ are constants computed only once, with their costs amortized over all key vectors. The remaining terms involving only key vectors or centroids, namely $||\mathbf{k} - \mathbf{c}_k||$, $\langle \mathbf{c}_q, \mathbf{k} \rangle$, and $\langle \mathbf{c}_q, \mathbf{c}_k \rangle$, can all be pre-calculated. Consequently, the core computational challenge is reduced to efficiently estimating the inner product between the two unit vectors, $\langle \mathbf{q}_c, \mathbf{k}_c \rangle$. Our subsequent discussion will focus on approximating this term with minimal computational overhead.

## A.2. Randomized Quantization Framework

Given that both $\mathbf{q}_c$ and $\mathbf{k}_c$ are unit vectors, we construct a base codebook from the $2^D$ vertices of a hypercube inscribed within a unit hypersphere, defined as $C := \left\{ \left( \pm \frac{1}{\sqrt{D}}, \ldots, \pm \frac{1}{\sqrt{D}} \right) \right\} \subset \mathbb{R}^D$. To handle arbitrary data distributions and mitigate alignment issues, we introduce a random rotation by applying a random orthogonal matrix $\mathbf{P} \in \mathbb{R}^{D \times D}$ to the base codebook. The resulting randomized codebook is $C_{\text{rand}} = \{\mathbf{Pc} \mid \mathbf{c} \in C\}$. Since $\mathbf{P}$ is orthogonal, all codewords in $C_{\text{rand}}$ remain unit vectors. For each normalized key vector $\mathbf{k}_c$, we find its nearest neighbor $\bar{\mathbf{k}}_c = \mathbf{P}\bar{\mathbf{c}}$ in the codebook $C_{\text{rand}}$. The search for the optimal codeword $\bar{\mathbf{c}}$ simplifies to maximizing an inner product:

$$\bar{\mathbf{c}} = \arg\min_{\mathbf{c} \in C} ||\mathbf{k}_c - \mathbf{Pc}||^2 \tag{7}$$

$$= \arg\min_{\mathbf{c} \in C} (||\mathbf{k}_c||^2 + ||\mathbf{Pc}||^2 - 2\langle \mathbf{k}_c, \mathbf{Pc} \rangle) \tag{8}$$

$$= \arg\min_{\mathbf{c} \in C} (2 - 2\langle \mathbf{P}^T \mathbf{k}_c, \mathbf{c} \rangle) \tag{9}$$

$$= \arg\max_{\mathbf{c} \in C} \langle \mathbf{P}^T \mathbf{k}_c, \mathbf{c} \rangle \tag{10}$$

The derivation relies on the property that $\mathbf{k}_c$ and $\mathbf{Pc}$ are unit vectors. Since all components of $\mathbf{c} \in C$ have the same absolute value, the inner product $\langle \mathbf{P}^T \mathbf{k}_c, \mathbf{c} \rangle$ is maximized when the sign of each component of $\mathbf{c}$ matches the sign of the corresponding component of $\mathbf{P}^T \mathbf{k}_c$. This insight enables highly efficient encoding. The quantized representation of $\mathbf{k}_c$ can be stored as a compact $D$-bit binary vector $\bar{\mathbf{c}}_b \in \{0, 1\}^D$, where the $i$-th bit is 1 if the $i$-th component of $\mathbf{P}^T \mathbf{k}_c$ is positive, and 0 otherwise. The original codeword $\bar{\mathbf{c}}$ can be losslessly recovered from $\bar{\mathbf{c}}_b$ via $\bar{\mathbf{c}} = (2\bar{\mathbf{c}}_b - \mathbf{1}_D)/\sqrt{D}$.

In summary, during the index construction phase, we compute and store two pieces of information for each key vector: (1) the quantized code $\bar{\mathbf{c}}_b$, a $D$-bit binary vector, and (2) a correction factor $\langle \bar{\mathbf{c}}', \mathbf{P}^T \mathbf{k}_c \rangle$, where $\bar{\mathbf{c}}' = (2\bar{\mathbf{c}}_b - \mathbf{1}_D)$, which will be explained in the following section.

## A.3. Online Inner Product Estimation

At query time, we leverage the pre-computed quantized representations to estimate the target inner product $\langle \mathbf{q}_c, \mathbf{k}_c \rangle$.

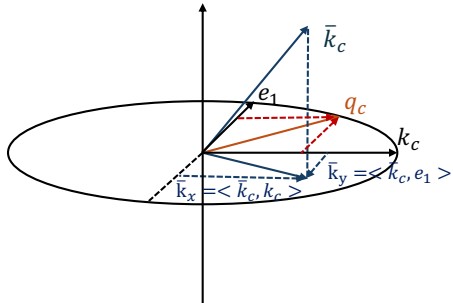

*Figure 6.* The geometric relationship of $\mathbf{q}_c, \mathbf{k}_c, \mathbf{e}_1$.

### A.3.1. AN UNBIASED ESTIMATOR

As shown in Figure 6, the geometric relationship between $\langle \mathbf{q}_c, \mathbf{k}_c \rangle$ and $\langle \bar{\mathbf{k}}_c, \mathbf{q}_c \rangle$ can be expressed by projecting $\mathbf{q}_c$ onto the subspace spanned by $\mathbf{k}_c$ and an orthogonal vector $\mathbf{e}_1$. This yields the relation:

$$\langle \bar{\mathbf{k}}_c, \mathbf{q}_c \rangle = \bar{k}_x \cdot \langle \mathbf{k}_c, \mathbf{q}_c \rangle + \bar{k}_y \cdot \sqrt{1 - \langle \mathbf{k}_c, \mathbf{q}_c \rangle^2} \tag{11}$$

$$= \langle \bar{\mathbf{k}}_c, \mathbf{k}_c \rangle \cdot \langle \mathbf{k}_c, \mathbf{q}_c \rangle + \langle \bar{\mathbf{k}}_c, \mathbf{e}_1 \rangle \sqrt{1 - \langle \mathbf{k}_c, \mathbf{q}_c \rangle^2} \tag{12}$$

where $\mathbf{e}_1 = \frac{\mathbf{k}_c - \langle \mathbf{k}_c, \mathbf{q}_c \rangle \mathbf{q}_c}{\sqrt{1 - \langle \mathbf{k}_c, \mathbf{q}_c \rangle^2}}$ is a unit vector orthogonal to $\mathbf{k}_c$ and coplanar with $\mathbf{q}_c$ and $\mathbf{k}_c$. Although a 2D projection is shown for intuition, this equation holds in any dimension $D \geq 2$. Solving this equation for $\langle \mathbf{q}_c, \mathbf{k}_c \rangle$ directly is complex. However, we observe that $\bar{\mathbf{k}}_c$ is determined by a random rotation, and $\mathbf{e}_1$ is defined in the orthogonal complement of $\mathbf{k}_c$. These vectors are thus statistically independent with respect to the random choice of $\mathbf{P}$. As previously mentioned, two high-dimensional random unit vectors tend to be orthogonal (Section 2.2), this leads to $\mathbb{E}[\langle \bar{\mathbf{k}}_c, \mathbf{e}_1 \rangle] = 0$. Taking the expectation of the equation above, we arrive at the following theorem.

**Theorem A.1.** *An unbiased estimator for $\langle \mathbf{k}_c, \mathbf{q}_c \rangle$ is given by:*

$$\mathbb{E}\left[ \frac{\langle \bar{\mathbf{k}}_c, \mathbf{q}_c \rangle}{\langle \bar{\mathbf{k}}_c, \mathbf{k}_c \rangle} \right] = \langle \mathbf{k}_c, \mathbf{q}_c \rangle \tag{13}$$

$$\left| \frac{\langle \bar{\mathbf{k}}_c, \mathbf{q}_c \rangle}{\langle \bar{\mathbf{k}}_c, \mathbf{k}_c \rangle} - \langle \mathbf{k}_c, \mathbf{q}_c \rangle \right| = O\left( \frac{1}{\sqrt{D}} \right) \text{ with high probability.} \tag{14}$$

The proof is provided in the Appendix A.4.1. With this result, the estimation problem reduces to computing the numerator $\langle \bar{\mathbf{k}}_c, \mathbf{q}_c \rangle$, as the denominator is the pre-computed correction factor.

### A.3.2. FAST COMPUTATION VIA LOW-PRECISION ARITHMETIC

The final step is to compute $\langle \bar{\mathbf{k}}_c, \mathbf{q}_c \rangle$ efficiently. By applying the property of orthogonal matrices, we have $\langle \bar{\mathbf{k}}_c, \mathbf{q}_c \rangle = \langle \mathbf{P}\bar{\mathbf{c}}, \mathbf{q}_c \rangle = \langle \bar{\mathbf{c}}, \mathbf{P}^T \mathbf{q}_c \rangle$. Let $\mathbf{q}' = \mathbf{P}^T \mathbf{q}_c$. The computation now involves a dot product between $\bar{\mathbf{c}}$ (a vector of $\pm 1/\sqrt{D}$) and the floating-point vector $\mathbf{q}'$. Since our overall method is an estimation, we can further accelerate this computation by quantizing $\mathbf{q}'$, provided that the introduced quantization error is asymptotically smaller than the existing estimation error of $O(1/\sqrt{D})$. The following theorem formalizes this.

**Theorem A.2.** *If we quantize $\mathbf{q}'$ to a $B_q$-bit integer vector $\bar{\mathbf{q}}$, selecting $B_q = \Omega(\log \log D)$ suffices to guarantee that $|\langle \bar{\mathbf{c}}, \mathbf{q}' \rangle - \langle \bar{\mathbf{c}}, \bar{\mathbf{q}} \rangle| = O\left( \frac{1}{\sqrt{D}} \right)$ with high probability.*

The proof is provided in the Appendix A.4.2. We then quantize $\mathbf{q}'$ using a uniform scalar quantizer. Let $[q_l, q_r]$ be the value range of $\mathbf{q}'$ and $\Delta = (q_r - q_l)/(2^{B_q} - 1)$. The quantized vector $\bar{\mathbf{q}}$ is obtained from a $B_q$-bit unsigned integer vector $\bar{\mathbf{q}}_u = \text{Round}((\mathbf{q}' - q_l)/\Delta)$ as $\bar{\mathbf{q}} = \Delta \cdot \bar{\mathbf{q}}_u + q_l \cdot \mathbf{1}_D$. The dot product is then approximated as:

$$\langle \bar{\mathbf{k}}_c, \mathbf{q}_c \rangle \approx \langle \bar{\mathbf{c}}, \bar{\mathbf{q}} \rangle \tag{15}$$

$$= \left\langle \frac{2\bar{\mathbf{c}}_b - \mathbf{1}_D}{\sqrt{D}}, \Delta \cdot \bar{\mathbf{q}}_u + q_l \cdot \mathbf{1}_D \right\rangle \tag{16}$$

$$= \frac{2\Delta}{\sqrt{D}} \langle \bar{\mathbf{c}}_b, \bar{\mathbf{q}}_u \rangle + \frac{2q_l}{\sqrt{D}} \sum_{i=1}^{D} \bar{\mathbf{c}}_b[i] - \frac{\Delta}{\sqrt{D}} \sum_{i=1}^{D} \bar{\mathbf{q}}_u[i] - \sqrt{D} \cdot q_l \tag{17}$$

Since the dimension D in LLM is generally greater than 128 (Yang et al., 2025; Dubey et al., 2024), a bit-width of $B_q = 4$ is sufficient to make the error from this quantization negligible. In the final expression, the first term is an inner product between a binary vector and an INT4 vector. The second term requires a popcount operation on the binary vector. The last two terms are scalars computed once per query. All these operations can be executed with exceptional efficiency on modern GPUs.

Since the estimated inner product we need to calculate is

$$est(\langle \mathbf{k}_c, \mathbf{q}_c \rangle) = \frac{\langle \bar{\mathbf{c}}, \bar{\mathbf{q}} \rangle}{\langle \bar{\mathbf{k}}_c, \mathbf{k}_c \rangle} = \frac{\langle \bar{\mathbf{c}}, \bar{\mathbf{q}} \rangle}{\langle \bar{\mathbf{c}}, \mathbf{P}^T \mathbf{k}_c \rangle} = \frac{\langle \bar{\mathbf{c}}', \bar{\mathbf{q}} \rangle}{\langle \bar{\mathbf{c}}', \mathbf{P}^T \mathbf{k}_c \rangle} \tag{18}$$

The last equation makes use of $\bar{\mathbf{c}}' = \sqrt{D}\bar{\mathbf{c}}$. Combining (16), (17) and (18), we can obtain our formula for computing $\langle \bar{\mathbf{c}}', \bar{\mathbf{q}} \rangle$:

$$\langle \bar{\mathbf{c}}', \bar{\mathbf{q}} \rangle = \langle 2\bar{\mathbf{c}}_b - \mathbf{1}_D, \Delta \cdot \bar{\mathbf{q}}_u + q_l \cdot \mathbf{1}_D \rangle \tag{19}$$

$$= 2\Delta \langle \bar{\mathbf{c}}_b, \bar{\mathbf{q}}_u \rangle + 2q_l \sum_{i=1}^{D} \bar{\mathbf{c}}_b[i] - \Delta \sum_{i=1}^{D} \bar{\mathbf{q}}_u[i] - D \cdot q_l \tag{20}$$

**Remark (Support for Top-$p$ Selection).** The unbiased nature of Eq. 13 is a distinct advantage over distance-based heuristics (e.g., K-means centroids). Because $E[\text{score}] = \langle k_c, q_c \rangle$, our proxy scores are numerically comparable to the true dot products. This allows us to approximate the Softmax denominator and employ threshold-based or cumulative-probability-based (Top-$p$) selection strategies, which are more robust to varying sparsity patterns than a fixed Top-$k$ approach.

## A.4. PROOF OF THEOREM in A

Our derivation below is inspired by RaBitQ (Gao & Long, 2024).

### A.4.1. PROOF OF THEOREM A.1

**Lemma A.3.** *(Vershynin, 2018) For a D-dimensional random vector* $\mathbf{x} = (\mathbf{x}[1], \mathbf{x}[2], \ldots, \mathbf{x}[D])$ *which follows the uniform distribution on the unit sphere, the probability density function of its every coordinate* $\mathbf{x}[1], \mathbf{x}[2], \ldots, \mathbf{x}[D]$ *is given as*

$$p_D(x) = \frac{\Gamma(\frac{D}{2})}{\sqrt{\pi}\Gamma(\frac{D-1}{2})} (1 - x^2)^{\frac{D-3}{2}}, x \in [-1, 1] \tag{31}$$

*where* $\Gamma(\cdot)$ *is the Gamma function. The tail bound is given as*

$$\mathbb{P}\left\{ |\mathbf{x}[i]| > \frac{t}{\sqrt{D}} \right\} \leq 2\exp(-c_0 t^2) \tag{32}$$

*where* $c_0$ *is a constant,* $i = 1, 2, \ldots, D$.

**Lemma A.4.** *Let* $\mathbf{k}_c$ *and* $\mathbf{e}_1$ *be two unit vectors, where* $\mathbf{k}_c \perp \mathbf{e}_1$. *Let P be a random orthogonal matrix drawn uniformly from the Haar measure, and let* $\mathcal{C}$ *be a deterministic codebook. Define* $\bar{\mathbf{c}} = \arg\max_{\mathbf{c} \in \mathcal{C}} \langle P^T \mathbf{k}_c, \mathbf{c} \rangle$ *and* $\bar{\mathbf{k}}_c = P\bar{\mathbf{c}}$. *Then, the joint distribution of the random vector* $(\langle \bar{\mathbf{k}}_c, \mathbf{k}_c \rangle, \langle \bar{\mathbf{k}}_c, \mathbf{e}_1 \rangle)$ *is identical to that of*

$$(\langle \bar{\mathbf{k}}_c, \mathbf{k}_c \rangle, \sqrt{1 - \langle \bar{\mathbf{k}}_c, \mathbf{k}_c \rangle^2} \cdot X_1) \tag{21}$$

*, where* $X_1$ *is independent of* $\langle \bar{\mathbf{k}}_c, \mathbf{k}_c \rangle$ *and* $X_1 \sim p_{D-1}$.

*Proof.* Let $c_{\text{par}} = \langle \bar{\mathbf{k}}_c, \mathbf{k}_c \rangle$. This is the scalar length of the parallel component (projection) of $\bar{\mathbf{k}}_c$ onto the axis of $\mathbf{k}_c$. The vector $\bar{\mathbf{k}}_c$ can be expressed as:

$$\bar{\mathbf{k}}_c = c_{\text{par}} \mathbf{k}_c + \bar{\mathbf{k}}_{c,\perp} \tag{22}$$

where $\bar{\mathbf{k}}_{c,\perp}$ is the component of $\bar{\mathbf{k}}_c$ in the orthogonal complement of $\mathbf{k}_c$, denoted $\mathbb{K}_c^\perp$. The norm of the perpendicular component is $\|\bar{\mathbf{k}}_{c,\perp}\| = \sqrt{1 - c_{\text{par}}^2}$. We can define a unit vector $\mathbf{v} \in \mathbb{K}_c^\perp$:

$$\mathbf{v} = \frac{\bar{\mathbf{k}}_{c,\perp}}{\|\bar{\mathbf{k}}_{c,\perp}\|} = \frac{\bar{\mathbf{k}}_c - c_{\text{par}} \mathbf{k}_c}{\sqrt{1 - c_{\text{par}}^2}} \tag{23}$$

Hence, $\bar{\mathbf{k}}_c$ can be precisely expressed as:

$$\bar{\mathbf{k}}_c = c_{\text{par}} \mathbf{k}_c + \sqrt{1 - c_{\text{par}}^2} \cdot \mathbf{v} \tag{24}$$

Using the above decomposition, we compute the two inner products in the lemma:

$$\langle \bar{\mathbf{k}}_c, \mathbf{k}_c \rangle = c_{\text{par}} \tag{25}$$

$$\langle \bar{\mathbf{k}}_c, \mathbf{e}_1 \rangle = c_{\text{par}} \langle \mathbf{k}_c, \mathbf{e}_1 \rangle + \sqrt{1 - c_{\text{par}}^2} \langle \mathbf{v}, \mathbf{e}_1 \rangle \tag{26}$$

$$= \sqrt{1 - c_{\text{par}}^2} \cdot \langle \mathbf{v}, \mathbf{e}_1 \rangle \tag{27}$$

Equations (27) is due to the fact that $\mathbf{k}_c \perp \mathbf{e}_1$. The proof reduces to showing that, conditional on the value of $c_{\text{par}}$, the vector $\mathbf{v}$ is uniformly distributed on the unit sphere of the subspace $\mathbf{k}_c^\perp$.

Consider an arbitrary orthogonal transformation $R$ that guarantees $R\mathbf{k}_c = \mathbf{k}_c$. Let $P' = RP$. Due to the left-invariance of the Haar measure (Haar, 1933), $P'$ is identically distributed to $P$. We use $P'$ to re-conduct the selection process of $\bar{\mathbf{c}}$:

$$\bar{\mathbf{c}}' = \arg\max_{\mathbf{c} \in \mathcal{C}} \langle (P')^T \mathbf{k}_c, \mathbf{c} \rangle \tag{28}$$

$$= \arg\max_{\mathbf{c} \in \mathcal{C}} \langle P^T R^T \mathbf{k}_c, \mathbf{c} \rangle \tag{29}$$

$$= \arg\max_{\mathbf{c} \in \mathcal{C}} \langle P^T \mathbf{k}_c, \mathbf{c} \rangle \tag{30}$$

Therefore, the selected codeword is the same, $\bar{\mathbf{c}}' = \bar{\mathbf{c}}$. The resulting vector is $\bar{\mathbf{k}}_c' = P'\bar{\mathbf{c}}' = (RP)\bar{\mathbf{c}} = R(P\bar{\mathbf{c}}) = R\bar{\mathbf{k}}_c$. Since $P$ and $P'$ have the same distribution, $\bar{\mathbf{k}}_c$ and $\bar{\mathbf{k}}_c'$ must also be identically distributed. This implies that the distribution of $\bar{\mathbf{k}}_c$ is rotationally symmetric about the $\mathbf{k}_c$ axis. This symmetry also holds for the conditional distribution. Conditioning on $\langle \bar{\mathbf{k}}_c, \mathbf{k}_c \rangle = c_{\text{par}}$, all the randomness in $\bar{\mathbf{k}}_c$ is captured by the unit vector $\mathbf{v}$. Therefore, the conditional distribution of $\mathbf{v}$ must be the uniform distribution on the unit sphere $S^{D-2}$ in the subspace $\mathbb{K}_c^\perp$.

Since the distribution of $\mathbf{v}$ is the same uniform for any given value of $c_{\text{par}}$, $\mathbf{v}$ and $c_{\text{par}}$ are statistically independent. It follows that $X_1 = \langle \mathbf{v}, \mathbf{e}_1 \rangle$ is also independent of $c_{\text{par}} = \langle \bar{\mathbf{k}}_c, \mathbf{k}_c \rangle$.

$\mathbf{v}$ is a uniformly random unit vector in the $(D-1)$-dimensional subspace $\mathbf{k}_c^\perp$, and $\mathbf{e}_1$ is a fixed unit vector in that same subspace. By definition (Vershynin, 2018), the distribution of their inner product $X_1 = \langle \mathbf{v}, \mathbf{e}_1 \rangle$ is $p_{D-1}$. $\square$

*Proof.* We first prove Equation (13). From Equation (12) we can deduce that:

$$\mathbb{E}\left[ \frac{\langle \bar{\mathbf{k}}_c, \mathbf{q}_c \rangle}{\langle \bar{\mathbf{k}}_c, \mathbf{k}_c \rangle} \right] = \mathbb{E}\left[ \langle \mathbf{k}_c, \mathbf{q}_c \rangle \right] + \mathbb{E}\left[ \frac{\langle \bar{\mathbf{k}}_c, \mathbf{e}_1 \rangle}{\langle \bar{\mathbf{k}}_c, \mathbf{k}_c \rangle} \sqrt{1 - \langle \mathbf{k}_c, \mathbf{q}_c \rangle^2} \right] \tag{31}$$

From Lemma A.4 we can obtain $\langle \bar{\mathbf{k}}_c, \mathbf{e}_1 \rangle \sim \sqrt{1 - \langle \bar{\mathbf{k}}_c, \mathbf{k}_c \rangle^2} \cdot X_1$. Now we substitute this into the expectation expression:

$$\mathbb{E}\left[ \frac{\langle \bar{\mathbf{k}}_c, \mathbf{q}_c \rangle}{\langle \bar{\mathbf{k}}_c, \mathbf{k}_c \rangle} \right] = \mathbb{E}\left[ \langle \mathbf{k}_c, \mathbf{q}_c \rangle \right] + \mathbb{E}\left[ \frac{\sqrt{1 - \langle \bar{\mathbf{k}}_c, \mathbf{k}_c \rangle^2}}{\langle \bar{\mathbf{k}}_c, \mathbf{k}_c \rangle} \sqrt{1 - \langle \mathbf{k}_c, \mathbf{q}_c \rangle^2} \cdot \mathbf{X}_1 \right] \tag{32}$$

$$= \mathbb{E}\left[ \langle \mathbf{k}_c, \mathbf{q}_c \rangle \right] + \mathbb{E}\left[ \frac{\sqrt{1 - \langle \bar{\mathbf{k}}_c, \mathbf{k}_c \rangle^2}}{\langle \bar{\mathbf{k}}_c, \mathbf{k}_c \rangle} \sqrt{1 - \langle \mathbf{k}_c, \mathbf{q}_c \rangle^2} \right] \mathbb{E}\left[ \mathbf{X}_1 \right] \tag{33}$$

Equation (33) holds because $X_1$ is independent of $\langle \bar{k}_c, k_c \rangle$ and $\langle k_c, q_c \rangle$ can be regarded as a constant for given $k_c, q_c$. Since $\mathbf{X}_1$ is a symmetrically distributed random variable, $\mathbb{E}[\mathbf{X}_1] = 0$, so we can conclude that:

$$\mathbb{E}\left[\frac{\langle \bar{\mathbf{k}}_c, \mathbf{q}_c \rangle}{\langle \bar{\mathbf{k}}_c, \mathbf{k}_c \rangle}\right] = \langle \mathbf{k}_c, \mathbf{q}_c \rangle \tag{34}$$

Next, we prove Equation (13). We can construct the following probability function.

$$P\left\{ \left| \frac{\langle \bar{\mathbf{k}}_c, \mathbf{q}_c \rangle}{\langle \bar{\mathbf{k}}_c, \mathbf{k}_c \rangle} - \langle \mathbf{k}_c, \mathbf{q}_c \rangle \right| > \frac{\sqrt{1 - \langle \bar{\mathbf{k}}_c, \mathbf{k}_c \rangle^2}}{\langle \bar{\mathbf{k}}_c, \mathbf{k}_c \rangle} \cdot \frac{t}{\sqrt{D-1}} \right\} \tag{35}$$

$$= P\left\{ \left| \frac{\langle \bar{\mathbf{k}}_c, \mathbf{e}_1 \rangle}{\langle \bar{\mathbf{k}}_c, \mathbf{k}_c \rangle} \sqrt{1 - \langle \mathbf{k}_c, \mathbf{q}_c \rangle^2} \right| > \frac{\sqrt{1 - \langle \bar{\mathbf{k}}_c, \mathbf{k}_c \rangle^2}}{\langle \bar{\mathbf{k}}_c, \mathbf{k}_c \rangle} \cdot \frac{t}{\sqrt{D-1}} \right\} \tag{36}$$

$$\leq P\left\{ \left| \frac{\langle \bar{\mathbf{k}}_c, \mathbf{e}_1 \rangle}{\langle \bar{\mathbf{k}}_c, \mathbf{k}_c \rangle} \right| > \frac{\sqrt{1 - \langle \bar{\mathbf{k}}_c, \mathbf{k}_c \rangle^2}}{\langle \bar{\mathbf{k}}_c, \mathbf{k}_c \rangle} \cdot \frac{t}{\sqrt{D-1}} \right\} \tag{37}$$

$$= P\left\{ \frac{\sqrt{1 - \langle \bar{\mathbf{k}}_c, \mathbf{k}_c \rangle^2}}{\langle \bar{\mathbf{k}}_c, \mathbf{k}_c \rangle} |\mathbf{X}_1| > \frac{\sqrt{1 - \langle \bar{\mathbf{k}}_c, \mathbf{k}_c \rangle^2}}{\langle \bar{\mathbf{k}}_c, \mathbf{k}_c \rangle} \cdot \frac{t}{\sqrt{D-1}} \right\} \tag{38}$$

$$= P\left\{ |\mathbf{X}_1| > \frac{t}{\sqrt{D-1}} \right\} \leq 2\exp(-c_0 t^2) \tag{39}$$

where (36) is due to Equation (12). (37) is due to bounding $\sqrt{1 - \langle \mathbf{k}_c, \mathbf{q}_c \rangle^2}$ by 1. (38) is due to lemma A.4. (39) is due to Lemma A.3. Therefore, we can conclude that $\left| \frac{\langle \bar{\mathbf{k}}_c, \mathbf{q}_c \rangle}{\langle \bar{\mathbf{k}}_c, \mathbf{k}_c \rangle} - \langle \mathbf{k}_c, \mathbf{q}_c \rangle \right| = O\left(\frac{1}{\sqrt{D}}\right)$ with high probability. $\qquad \square$

### A.4.2. PROOF OF THEOREM A.2

We provide the proof for Theorem A.2, which states that selecting $B_q = \Omega(\log\log D)$ suffices to guarantee that $|\langle \bar{\mathbf{c}}, \mathbf{q}' \rangle - \langle \bar{\mathbf{c}}, \bar{\mathbf{q}} \rangle| = O\left(\frac{1}{\sqrt{D}}\right)$ with high probability. The proof is structured into three main steps. First, we bound the quantization error in terms of the quantization step size $\Delta$. Second, we bound $\Delta$ itself. Finally, we combine these bounds to determine the required bit-width $B_q$.

**Lemma A.5.** *If we quantize $\mathbf{q}'$ to a $B_q$-bit vector $\bar{\mathbf{q}}$ using randomized uniform scalar quantization, let $[q_l, q_r]$ be the value range of $\mathbf{q}'$ and $\Delta = (q_r - q_l)/(2^{B_q} - 1)$. The quantization error $|\langle \bar{\mathbf{c}}, \mathbf{q}' \rangle - \langle \bar{\mathbf{c}}, \bar{\mathbf{q}} \rangle| = O(\Delta)$ with high probability.*

*Proof.* We expand the quantization error term:

$$|\langle \bar{\mathbf{c}}, \mathbf{q}' \rangle - \langle \bar{\mathbf{c}}, \bar{\mathbf{q}} \rangle| = |\langle \bar{\mathbf{c}}, \mathbf{q}' - \bar{\mathbf{q}} \rangle| = \left| \sum_{i=1}^{D} \bar{\mathbf{c}}[i] \cdot (\mathbf{q}'[i] - \bar{\mathbf{q}}[i]) \right| \tag{40}$$

Let $\epsilon_i = \mathbf{q}'[i] - \bar{\mathbf{q}}[i]$ be the quantization error for the $i$-th component. By definition of rounding, we have $|\epsilon_i| \leq \Delta$. However, to obtain a sharper bound, we consider randomized rounding. Because we are using uniform quantization, this makes the expected quantization error for a single component zero: $\mathbb{E}[\epsilon_i] = 0$.

The term inside the summation is $X_i = \bar{\mathbf{c}}[i] \cdot (\mathbf{q}'[i] - \bar{\mathbf{q}}[i])$. Recall that $\bar{\mathbf{c}}[i] = \pm 1/\sqrt{D}$. The quantization error $\epsilon_i$ is bounded by $[-\Delta, \Delta]$. Thus, each random variable $X_i$ is bounded by $[-\Delta/\sqrt{D}, \Delta/\sqrt{D}]$. Crucially, the randomization in the quantization of each component is independent. Therefore, $X_1, \ldots, X_D$ are independent random variables with $\mathbb{E}[X_i] = 0$.

we apply Hoeffding's inequality (Vershynin, 2018), which states that for a sum of $n$ independent bounded variables $S_n = \sum X_i$, $\mathbb{P}(|S_n - \mathbb{E}[S_n]| \geq t) \leq 2\exp(-2t^2 / \sum(b_i - a_i)^2)$.

In our case, we note that $a_i = -\Delta/\sqrt{D}, b_i = +\Delta/\sqrt{D}, \mathbb{E}[S_n] = 0$. Therefore, we can obtain the following conclusion.

$$\mathbb{P}\left(\left|\sum_{i=1}^{D} \bar{\mathbf{c}}[i] \cdot (\mathbf{q}'[i] - \bar{\mathbf{q}}[i])\right| \geq t\right) \leq 2\exp\left(-\frac{t^2}{2\Delta^2}\right) \tag{41}$$

$$\mathbb{P}\left(\left|\sum_{i=1}^{D} \bar{\mathbf{c}}[i] \cdot (\mathbf{q}'[i] - \bar{\mathbf{q}}[i])\right| \geq \Delta u\right) \leq 2\exp\left(-\frac{u^2}{2}\right) \tag{42}$$

where 42 is due to the setting $u = t/\Delta$. This shows that the error $|\langle \bar{\mathbf{c}}, \mathbf{q}'\rangle - \langle \bar{\mathbf{c}}, \bar{\mathbf{q}}\rangle|$ is bounded by $O(\Delta)$ with high probability. $\qquad\square$

**Lemma A.6.** *The quantization step size* $\Delta = (q_r - q_l)/(2^{B_q} - 1) = O\left(\sqrt{\frac{\log D}{D}}/2^{B_q}\right)$ *with high probability.*

*Proof.* The step size $\Delta$ depends on the value range $[q_l, q_r]$ of the vector $\mathbf{q}' = P^T\mathbf{q}^c$. Since $\mathbf{q}^c$ is a fixed unit vector and $P$ is a random orthogonal matrix, $\mathbf{q}'$ is a uniformly random vector on the unit sphere $S^{D-1}$. The range is given by $q_r - q_l = \max_i \mathbf{q}'[i] - \min_i \mathbf{q}'[i] \leq 2\max_i |\mathbf{q}'[i]|$. We need to bound the maximum absolute value of any component of a random unit vector.

From Lemma A.3, for a random unit vector, the tail bound for any component $\mathbf{q}'[i]$ is:

$$\mathbb{P}\left(|\mathbf{q}'[i]| > \frac{t}{\sqrt{D}}\right) \leq 2\exp(-c_0 t^2) \tag{43}$$

We can use the union bound to bound the maximum of all $D$ components:

$$\mathbb{P}\left(\max_i |\mathbf{q}'[i]| > \frac{t}{\sqrt{D}}\right) \tag{44}$$

$$= \mathbb{P}\left(\exists 1 \leq i \geq D, |\mathbf{q}'[i]| > \frac{t}{\sqrt{D}}\right) \tag{45}$$

$$\leq \sum_{i=1}^{D} \mathbb{P}\left(|\mathbf{q}'[i]| > \frac{t}{\sqrt{D}}\right) \tag{46}$$

$$\leq D \cdot 2\exp(-c_0 t^2) \tag{47}$$

To make this probability small, we need $D \cdot 2\exp(-c_0 t^2)$ to be small. Let's set $t = \sqrt{(\log D + u)/c_0}$ for some constant $u > 0$.

$$\mathbb{P}\left(\max_i |\mathbf{q}'[i]| > \sqrt{\frac{\log D + u}{c_0 D}}\right) \leq 2D\exp(-c_0 \frac{\log D + u}{c_0}) \tag{48}$$

$$= 2D\exp(-\log D - u) \tag{49}$$

$$= 2D \cdot D^{-1} \cdot \exp(-u) = 2\exp(-u) \tag{50}$$

This implies that with high probability, $\max_i |\mathbf{q}'[i]| = O\left(\sqrt{\frac{\log D}{D}}\right)$. Therefore, the range $q_r - q_l = O\left(\sqrt{\frac{\log D}{D}}\right)$, and the step size is:

$$\Delta = \frac{q_r - q_l}{2^{B_q} - 1} = O\left(\sqrt{\frac{\log D}{D}}/2^{B_q}\right) \quad \text{with high probability} \tag{51}$$

$$\square$$

**Proof of Theorem A.2.** By combining Lemma A.5 and Lemma A.6, the total quantization error is:

$$|\langle \bar{\mathbf{c}}, \mathbf{q}'\rangle - \langle \bar{\mathbf{c}}, \bar{\mathbf{q}}\rangle| = O(\Delta) = O\left(\sqrt{\frac{\log D}{D}}/2^{B_q}\right) \quad \text{with high probability} \tag{52}$$

Our goal is to make this quantization error asymptotically smaller than or equal to the main estimation error of the framework, which is $O(1/\sqrt{D})$.

$$O\left(\sqrt{\frac{\log D}{D}}/2^{B_q}\right) \leq O\left(\frac{1}{\sqrt{D}}\right) \tag{53}$$

$$\frac{\sqrt{\log D}}{2^{B_q}} \leq O(1) \tag{54}$$

$$2^{B_q} \geq \Omega(\sqrt{\log D}) \tag{55}$$

$$B_q \geq \Omega(\log(\sqrt{\log D})) = \Omega(\log \log D) \tag{56}$$

where (53)-(56) are obtained by simple algebraic manipulations. Thus, selecting $B_q = \Omega(\log \log D)$ is sufficient to ensure the quantization error $|\langle \bar{\mathbf{c}}, \mathbf{q}' \rangle - \langle \bar{\mathbf{c}}, \bar{\mathbf{q}} \rangle|$ is at most $O(1/\sqrt{D})$ with high probability. This completes the proof. In practice, as $\log \log D$ grows very slowly, a small constant like $B_q = 4$ is empirically sufficient for typical dimensions. $\square$

## B. Kernel Optimization Details

### B.1. Top-$p$ Operator Kernel Optimization Details

In this section, we elaborate on the low-level hardware optimizations implemented in our Top-$p$ operator to minimize latency and maximize memory bandwidth utilization on GPUs.

**Vectorized Memory Access & Storage Reuse.** Since the Top-$p$ operation is memory-bound, effective bandwidth utilization is critical. We employ vectorized load instructions aligned with the GPU memory bus width, coupled with a dynamic dispatch mechanism that selects the optimal vector size at runtime. Furthermore, we implement a union-based storage strategy in shared memory. This allows the same memory region to be strictly reused for multiple purposes—serving as reduction buffers during computation and as broadcasting channels for parameter updates—thereby significantly improving occupancy and reducing the shared memory footprint.

**Hierarchical Reduction & Scheduling.** To minimize synchronization overhead, we adopt a two-stage hierarchical reduction strategy. Threads first accumulate partial results in local registers before performing warp-level tree-based reductions. This approach substantially reduces bank conflicts and traffic in shared memory compared to standard atomic operations. Additionally, the kernel utilizes architecture-aware configurations to dynamically select the optimal thread block size, ensuring high utilization across different generations of GPU architectures.

**Instruction-Level Parallelism (ILP).** At the instruction level, we heavily utilize loop unrolling strategies. By explicitly unrolling loops, we reduce loop control overhead (branching and index arithmetic) and expose more independent instructions to the compiler. This allows for better instruction scheduling and maximizes the throughput of the computational pipeline.

## C. Complexity Analysis

In this section, we provide a rigorous theoretical analysis of the time and space complexity of RaBitQCache compared to the standard Full Attention mechanism. We analyze the complexity with respect to the sequence length $L$, the number of attention heads $N_h$, the head dimension $D$, and the selected subset size $K_{select}$ .

### C.1. Space Complexity Analysis

We analyze the storage overhead introduced by RaBitQCache compared to a standard KVCache implementation. Let $L$ be the sequence length, $N_h$ be the number of attention heads, $D$ be the head dimension, and $B_{data}$ be the number of bytes per element in the original KVCache (e.g., $B_{data} = 2$ for FP16). **Standard KVCache:** The standard KVCache stores both Key and Value matrices. The total memory consumption is:

$$\mathcal{S}_{KV} = 2 \cdot L \cdot N_h \cdot D \cdot B_{data} \text{ bytes} \tag{57}$$

**RaBitQCache Index Overhead:** RaBitQCache constructs a lightweight retrieval index consisting of two components:

1. **Binary Index ($\bar{C}_b$):** Stores a 1-bit quantized representation for each dimension of the Key vector. The size is $L \cdot N_h \cdot D$ bits, or equivalently $L \cdot N_h \cdot \frac{D}{8}$ bytes.

2. **Correction Factor ($\alpha$):** Stores one scalar per token per head to ensure unbiased estimation. Let $B_\alpha$ be the bytes required for this scalar (typically stored in FP16, so $B_\alpha = 2$). The size is $L \cdot N_h \cdot B_\alpha$ bytes.

The extra space complexity introduced by RaBitQCache is:

$$\mathcal{S}_{Index} = L \cdot N_h \cdot \left( \frac{D}{8} + B_\alpha \right) \text{ bytes} \tag{58}$$

**Overhead Ratio:** To quantify the cost, we define the overhead ratio $\eta$ as the size of the index relative to the full KVCache:

$$\eta = \frac{\mathcal{S}_{Index}}{\mathcal{S}_{KV}} = \frac{L \cdot N_h \cdot (D/8 + B_\alpha)}{2 \cdot L \cdot N_h \cdot D \cdot B_{data}} = \frac{D/8 + B_\alpha}{2 \cdot D \cdot B_{data}} \tag{59}$$

**Case Study (FP16):** Assuming a typical configuration where the KVCache uses FP16 precision ($B_{data} = 2$), the correction factor is FP16 ($B_\alpha = 2$), and the head dimension $D = 128$:

$$\eta_{FP16} = \frac{128/8 + 2}{2 \cdot 128 \cdot 2} = \frac{16 + 2}{512} = \frac{18}{512} \approx 3.5\% \tag{60}$$

**Analysis:** The RaBitQCache index introduces a negligible memory overhead (approx. 3.5% for FP16). This extreme compactness allows the index to reside permanently in high-bandwidth GPU memory with minimal impact on capacity. Crucially, this small footprint decouples the retrieval logic from the bulk data storage, making it feasible to offload the massive full-precision $\mathcal{S}_{KV}$ to cheaper host memory or storage while retaining high-performance retrieval capabilities on the GPU.

### C.2. Time Complexity

We analyze the computational complexity for the Prefill and Decoding phases separately.

**Prefill Phase.** In the standard attention mechanism, the model computes attention scores for all token pairs, leading to a quadratic complexity of $\mathcal{O}(N_h \cdot L^2 \cdot D)$. For RaBitQCache, the index construction involves three steps for each token: (1) random rotation ($\mathcal{O}(N_h \cdot L \cdot D^2)$), (2) binary quantization to generate the index $\bar{C}_b$ ($\mathcal{O}(N_h \cdot L \cdot D)$), and (3) computing the scalar correction factor $\alpha$ ($\mathcal{O}(D)$). Therefore, the total indexing complexity is $\mathcal{O}(N_h \cdot L \cdot D^2)$.

Since the head dimension $D$ is constant and $D \ll L$ in long-context scenarios, this linear overhead is asymptotically negligible compared to the quadratic cost of the standard attention computation. Furthermore, in our system implementation, these element-wise and matrix-vector operations are pipelined and effectively masked by the computationally intensive attention calculation and memory movement of the prefill phase.

**Decoding Phase (Per Token).** During the decoding step $t$, the system generates one token. In the standard attention mechanism, the model computes the inner product of the new query with all cached keys, resulting in a complexity of $\mathcal{O}(N_h \cdot L \cdot D)$ using floating-point operations. However, the primary bottleneck is typically memory bandwidth, constrained by the latency of loading the full KV Cache.

In contrast, RaBitQCache scans the binary index to estimate scores. Although the theoretical complexity remains linear at $\mathcal{O}(N_h \cdot L \cdot D)$, it utilizes the efficient INT4_dot_Binary GEMV operator, which significantly accelerates execution speed compared to high-precision GEMV. Subsequently, our optimized Top-$p$ kernel performs candidate selection in $\mathcal{O}(L)$. Finally, the precise attention computation is restricted to the selected subset $K_{sel}$ (where $K_{sel} \ll L$) and the local sliding window $W$, reducing the computational cost to $\mathcal{O}(N_h \cdot K_{sel} \cdot D)$.

**Summary:** RaBitQCache maintains a linear scan complexity but significantly reduces the constant factor and, more importantly, the memory I/O volume. By filtering irrelevant tokens using the lightweight index, the expensive full-precision attention computation is reduced from $\mathcal{O}(L \cdot D)$ to $\mathcal{O}(K_{sel} \cdot D)$, ensuring low latency even as the context length $L$ scales to hundreds of thousands.

## D. Full Results on Longbench

Please see details in Table 6 and 7. The recall rate is the attention score divided by the attention score of Full Attention.

*Table 6.* Full results on Longbench. The table spans the full width of the page to accommodate all datasets. *MF-en* is abbreviated for MultiFieldQA-en, *NarrQA* for NarrativeQA, *Hotpot* for HotpotQA, *2Wiki* for 2WikiMQA, *Musiq* for Musique, *GovRep* for GovReport, *MultiN* for MultiNews, *Trivia* for TriviaQA, *PR-en* for PassageRetrieval-en, *Repo* for Repobench-P. The best results for each model (excluding Full and Oracle) are marked in **bold**.

| Method | Budget | Single-Doc QA | | | Multi-Doc QA | | | Summarization | | | Few | Syn | Code | | Avg |
|---|---|---|---|---|---|---|---|---|---|---|---|---|---|---|---|
| | | Qasper | MF-en | NarrQA | Hotpot | 2Wiki | Musiq | GovRep | QMSum | MultiN | Trivia | PR-en | LCC | Repo | |
| *Longchat-7B-v1.5-32k* | | | | | | | | | | | | | | | |
| Full | - | 28.66 | 41.51 | 21.35 | 31.18 | 23.40 | 13.11 | 31.07 | 22.77 | 26.25 | 83.55 | 30.00 | 54.35 | 57.75 | 35.78 |
| Oracle | 1024 | 30.37 | 41.18 | 19.94 | 31.8 | 22.13 | 12.78 | 30.84 | 23.14 | 26.41 | 84.35 | 29 | 55.71 | 58.34 | 35.84 |
| RaBitQCache | $p$=0.95 | 30.17 | 42.78 | **21.07** | 32.36 | 23.07 | **14.50** | **31.60** | 23.29 | 26.68 | 84.45 | 28.00 | 54.86 | 57.9 | 36.21 |
| Quest | 256 | 25.66 | 32.39 | 13.49 | 21.72 | **24.74** | 8.67 | 22.64 | 20.82 | 24.93 | 70.29 | 32.92 | 50.68 | 47.50 | 30.50 |
| | 1024 | **30.41** | 41.93 | 17.99 | 30.34 | 23.89 | 9.69 | 29.93 | 22.41 | 26.30 | 83.91 | **34.00** | 52.47 | 55.69 | 35.30 |
| | 4096 | 29.67 | **43.04** | 20.05 | 33.04 | 23.56 | 13.56 | 31.40 | 22.95 | 26.64 | **84.81** | 30.50 | 54.40 | 57.98 | **36.28** |
| | 8192 | 28.73 | 42.16 | 20.87 | **33.14** | 23.55 | 13.21 | 31.14 | 23.11 | 26.29 | 83.96 | 31.00 | 54.69 | 57.63 | 36.11 |
| DS | 256 | 3.28 | 3.32 | 0.64 | 0.21 | 1.02 | 0.18 | 1.46 | 2.06 | 6.68 | 1.95 | 1.33 | 17.20 | 9.38 | 3.75 |
| | 1024 | 11.86 | 15.04 | 0.72 | 1.52 | 5.61 | 0.27 | 4.43 | 2.65 | 21.14 | 11.10 | 0.82 | 34.73 | 13.88 | 9.52 |
| | 4096 | 28.20 | 37.08 | 3.21 | 9.33 | 20.32 | 1.15 | 18.80 | 12.06 | 26.08 | 33.55 | 1.92 | 52.16 | 32.77 | 21.28 |
| | 8192 | 28.56 | 41.31 | 10.65 | 28.58 | 23.31 | 8.81 | 27.65 | 21.08 | 26.34 | 70.81 | 32.00 | 54.19 | 50.29 | 32.58 |
| SparQ | Ratio=0.25 | 29.24 | 40.90 | 20.60 | 31.25 | 22.65 | 13.07 | 30.73 | 22.90 | 26.42 | 84.30 | 30.50 | **55.90** | **58.39** | 35.91 |
| *LLaMA-3.1-8B-Instruct* | | | | | | | | | | | | | | | |
| Full | - | 44.78 | 53.39 | 29.09 | 55.95 | 43.45 | 27.92 | 34.78 | 25.18 | 27.58 | 91.17 | 99.50 | 64.76 | 59.97 | 50.58 |
| Oracle | 1024 | 45.39 | 53.10 | 29.58 | 56.16 | 38.57 | 25.47 | 33.73 | 25.14 | 27.47 | 91.97 | 99.50 | 65.53 | 62.37 | 50.31 |
| RabitQCache | - | 46.29 | 52.97 | 29.65 | 56.16 | 41.07 | **27.87** | **34.96** | 25.25 | 27.24 | 91.67 | **99.50** | 64.74 | 60.83 | **50.63** |
| Quest | 256 | 25.12 | 37.77 | 10.27 | 33.13 | 26.67 | 10.54 | 26.16 | 19.48 | 27.27 | 63.81 | 93.00 | 55.59 | 48.48 | 36.71 |
| | 1024 | 39.05 | 50.52 | 25.94 | 46.26 | 39.34 | 20.78 | 33.24 | 23.67 | 27.50 | 79.46 | **99.50** | 60.76 | 58.75 | 46.52 |
| | 4096 | 46.83 | 51.31 | 28.43 | 51.98 | 36.72 | 24.37 | 34.77 | 25.14 | 27.46 | 89.82 | **99.50** | 64.22 | 59.62 | 49.24 |
| | 8192 | **46.83** | 52.90 | 28.55 | 54.82 | **44.43** | 25.27 | 34.85 | **25.51** | 27.35 | **91.73** | **99.50** | 64.05 | 59.86 | 50.43 |
| DS | 256 | 44.78 | 51.79 | 28.36 | 46.95 | 39.76 | 26.04 | 34.13 | 24.80 | 27.32 | 89.83 | **99.50** | 63.27 | 61.21 | 49.06 |
| | 1024 | 45.71 | 52.58 | 29.51 | 54.61 | 40.68 | 25.37 | 34.15 | 24.83 | 27.32 | 91.58 | **99.50** | 66.10 | **61.72** | 50.28 |
| | 4096 | 44.81 | **53.40** | 29.38 | 54.81 | 43.06 | 25.43 | 34.38 | 25.25 | 27.49 | 91.48 | **99.50** | 64.81 | 61.46 | 50.40 |
| | 8192 | 45.21 | 53.06 | 29.44 | 55.27 | 43.26 | 26.43 | 34.76 | 25.06 | **27.57** | 91.64 | **99.50** | 64.89 | 60.04 | 50.47 |
| SparQ | Ratio=0.25 | 44.10 | 53.30 | **29.87** | **56.22** | 38.46 | 25.23 | 34.05 | 24.99 | 27.34 | 91.48 | **99.50** | **66.20** | 61.27 | 50.15 |

# E. Additional Accuracy Experiments

In this section, we report the detailed per-task results corresponding to the summarized tables in Section 5.2.

### E.1. Comparison with Additional Sparse Attention Baselines on LongBench

Table 8 reports the full per-task results of RaBitQCache against MagicPIG, PyramidKV, SnapKV, PQCache, and KIVI on LongBench (LLaMA-3.1-8B-Instruct). All baselines use their official default configurations, as reported in Section 5.2. RaBitQCache uses $p = 0.95$.

### E.2. LongBench Results on LLaMA-3.1-70B-Instruct

Table 9 reports the per-task LongBench scores on LLaMA-3.1-70B-Instruct. RaBitQCache uses $p = 0.9$, and Quest uses a 1024-token budget. We additionally report the realized average token budget and the per-task attention recall for each sparse baseline. The results clearly show that, in contrast to fixed-budget Top-$k$ baselines, RaBitQCache adaptively allocates the budget across tasks (e.g., 856.9 tokens on multi_news vs. 3550.5 tokens on narrativeqa), which is the key advantage of Top-$p$ retrieval.

### E.3. Synthetic Long-Context Evaluation on RULER

Table 10 reports the full RULER results across context lengths from 8K to 64K on LLaMA-3.1-8B-Instruct. DS and Quest use a fixed 4096-token budget, SparQ uses its default setting, and RaBitQCache uses adaptive Top-$p$ with $p = 0.9$. The last column reports the realized average token budget of RaBitQCache, which scales naturally with context length. RaBitQCache

uses substantially fewer tokens than the fixed-budget baselines while maintaining competitive or superior accuracy. Notably, at 16K our method even surpasses Full Attention (80.3 vs. 78.6), suggesting that the adaptive sparsity may also act as a mild regularizer against irrelevant context.

### E.4. Mathematical Reasoning on GSM8K

GSM8K stresses sparse attention methods in the long-output regime, since the model must generate a multi-step chain of thought. We evaluate on LLaMA-3.1-8B-Instruct. Full Attention uses an average budget of 776.5 tokens; for fixed-budget baselines we set DS and Quest to 256 tokens (a strict setting that mimics aggressive eviction); SparQ uses its default. As shown in Table 11, RaBitQCache matches the accuracy of DS and SparQ (0.77) while attaining the highest attention recall (0.888) among all sparse baselines, suggesting that even when the output length is large, our proxy score remains a faithful estimator of the true attention.

### E.5. Centroid Re-centering Ablation

Table 12 reports per-task LongBench scores on LLaMA-3.1-8B-Instruct, comparing RaBitQCache against a variant that omits the centroid re-centering step described in Section 3.1. The average score drops from 50.63 to 50.25, with most tasks showing only minor differences. This confirms that, although re-centering is theoretically required to satisfy the unit-hypersphere uniformity assumption underlying our error bound, its practical impact under typical long-context workloads is modest.

*Table 7.* Statistics of Token **Budget** (Bud.) and Retrieval **Recall** of attention score (Rec.) in percentage (%) on LongBench.

| Method | Metric | Single-Doc QA | | | Multi-Doc QA | | | Summarization | | | Few | Syn | Code | | Avg |
|---|---|---|---|---|---|---|---|---|---|---|---|---|---|---|---|
| | | Qasper | MF-en | NarrQA | Hotpot | 2Wiki | Musiq | GovRep | QMSum | MultiN | Trivia | PR-en | LCC | Repo | |
| *Longchat-7B-v1.5-32k* | | | | | | | | | | | | | | | |
| Full (None) | Bud. | 100.00 | 100.00 | 100.00 | 100.00 | 100.00 | 100.00 | 100.00 | 100.00 | 100.00 | 100.00 | 100.00 | 100.00 | 100.00 | 100.00 |
| | Rec. | 100.00 | 100.00 | 100.00 | 100.00 | 100.00 | 100.00 | 100.00 | 100.00 | 100.00 | 100.00 | 100.00 | 100.00 | 100.00 | 100.00 |
| Oracle(1k) | Bud. | 17.55 | 12.55 | 4.14 | 6.67 | 12.02 | 5.51 | 8.42 | 6.43 | 29.88 | 7.27 | 7.05 | 23.79 | 7.13 | 11.42 |
| | Rec. | 91.02 | 89.54 | 81.45 | 84.82 | 89.05 | 83.82 | 88.29 | 86.64 | 95.75 | 89.62 | 88.52 | 93.93 | 88.52 | 89.30 |
| RaBitQCache | Bud. | 24.33 | 20.15 | 11.91 | 14.77 | 19.54 | 13.45 | 16.04 | 14.77 | 27.04 | 13.23 | 13.83 | 23.79 | 12.50 | 17.33 |
| | Rec. | 91.38 | 91.04 | 86.02 | 89.15 | 90.73 | 88.09 | 89.19 | 89.15 | 91.53 | 89.56 | 90.93 | 92.00 | 89.24 | 89.85 |
| SparQ | Bud. | 25.00 | 25.00 | 25.00 | 25.00 | 25.00 | 25.00 | 25.00 | 25.00 | 24.99 | 25.00 | 25.00 | 25.01 | 25.00 | 25.00 |
| | Rec. | 92.01 | 92.09 | 94.04 | 93.37 | 92.01 | 94.19 | 93.51 | 94.51 | 90.42 | 94.92 | 94.97 | 90.02 | 94.62 | 93.13 |
| Quest (256) | Bud. | 4.35 | 3.11 | 1.03 | 1.66 | 2.98 | 1.37 | 2.09 | 1.60 | 7.55 | 1.81 | 1.75 | 5.90 | 1.77 | 2.84 |
| | Rec. | 28.35 | 29.48 | 19.38 | 22.50 | 29.08 | 20.91 | 24.71 | 21.42 | 40.92 | 29.12 | 27.79 | 34.27 | 26.71 | 27.28 |
| Quest (1k) | Bud. | 17.49 | 12.51 | 4.13 | 6.64 | 11.98 | 5.49 | 8.39 | 6.40 | 29.77 | 7.24 | 7.02 | 23.70 | 7.11 | 11.38 |
| | Rec. | 61.31 | 60.25 | 43.40 | 49.58 | 60.03 | 46.71 | 55.71 | 48.97 | 77.72 | 62.54 | 59.77 | 71.96 | 58.62 | 58.12 |
| Quest (4k) | Bud. | 66.66 | 47.17 | 16.55 | 26.51 | 46.53 | 22.01 | 33.37 | 25.64 | 81.37 | 28.35 | 28.15 | 73.00 | 28.48 | 40.29 |
| | Rec. | 93.57 | 89.47 | 71.54 | 79.74 | 89.23 | 76.23 | 85.71 | 79.25 | 98.44 | 87.74 | 83.75 | 97.50 | 86.41 | 86.05 |
| Quest (8k) | Bud. | 93.16 | 80.21 | 33.11 | 51.59 | 80.27 | 44.03 | 61.23 | 50.35 | 94.67 | 52.51 | 56.33 | 89.11 | 52.90 | 64.57 |
| | Rec. | 99.45 | 97.72 | 85.01 | 91.76 | 97.92 | 89.03 | 95.27 | 91.59 | 99.82 | 95.25 | 93.68 | 99.58 | 95.10 | 94.71 |
| DS (256) | Bud. | 4.39 | 3.14 | 1.04 | 1.67 | 3.01 | 1.38 | 2.10 | 1.61 | 7.61 | 1.82 | 1.76 | 5.95 | 1.78 | 2.86 |
| | Rec. | 77.66 | 46.76 | 1.99 | 0.05 | 0.22 | 0.01 | 0.10 | 0.02 | 2.63 | 0.16 | 0.02 | 0.85 | 0.07 | 10.04 |
| DS (1k) | Bud. | 17.55 | 12.55 | 4.14 | 6.67 | 12.02 | 5.51 | 8.42 | 6.43 | 29.88 | 7.27 | 7.05 | 23.79 | 7.13 | 11.42 |
| | Rec. | 97.16 | 86.05 | 19.99 | 0.59 | 3.65 | 0.10 | 1.00 | 0.20 | 29.37 | 1.97 | 0.16 | 14.12 | 0.65 | 20.38 |
| DS (4k) | Bud. | 66.73 | 47.22 | 16.57 | 26.54 | 46.58 | 22.03 | 33.41 | 25.68 | 81.41 | 28.39 | 28.19 | 73.04 | 28.52 | 40.33 |
| | Rec. | 100.00 | 100.00 | 95.81 | 10.61 | 48.11 | 1.30 | 27.09 | 9.83 | 93.02 | 23.02 | 2.20 | 87.32 | 22.98 | 47.79 |
| DS (8k) | Bud. | 93.17 | 80.25 | 33.14 | 51.62 | 80.29 | 44.06 | 61.26 | 50.38 | 94.67 | 52.54 | 56.37 | 89.12 | 52.92 | 64.60 |
| | Rec. | 100.00 | 100.00 | 100.00 | 35.86 | 85.52 | 12.88 | 64.32 | 40.95 | 98.46 | 48.83 | 40.36 | 96.42 | 53.00 | 67.43 |
| *LLaMA-3.1-8B-Instruct* | | | | | | | | | | | | | | | |
| Full (None) | Bud. | 100.00 | 100.00 | 100.00 | 100.00 | 100.00 | 100.00 | 100.00 | 100.00 | 100.00 | 100.00 | 100.00 | 100.00 | 100.00 | 100.00 |
| | Rec. | 100.00 | 100.00 | 100.00 | 100.00 | 100.00 | 100.00 | 100.00 | 100.00 | 100.00 | 100.00 | 100.00 | 100.00 | 100.00 | 100.00 |
| Oracle(1k) | Bud. | 20.07 | 14.73 | 4.30 | 7.96 | 14.27 | 6.55 | 9.95 | 7.33 | 34.93 | 8.69 | 8.18 | 32.00 | 9.49 | 13.73 |
| | Rec. | 89.35 | 87.70 | 74.92 | 81.97 | 87.89 | 78.72 | 84.25 | 81.50 | 95.46 | 88.70 | 89.84 | 94.87 | 84.52 | 86.90 |
| RaBitQCache | Bud. | 26.62 | 23.60 | 15.79 | 18.08 | 21.79 | 17.21 | 22.15 | 20.17 | 32.67 | 18.37 | 15.52 | 28.62 | 18.33 | 21.45 |
| | Rec. | 91.28 | 90.88 | 89.55 | 90.11 | 90.84 | 89.10 | 91.00 | 90.23 | 92.60 | 91.62 | 90.58 | 91.60 | 90.11 | 90.73 |
| SparQ | Bud. | 25.01 | 25.00 | 25.00 | 25.00 | 25.00 | 25.00 | 25.00 | 25.00 | 25.00 | 25.00 | 25.00 | 25.01 | 25.00 | 25.00 |
| | Rec. | 86.93 | 88.41 | 90.45 | 89.91 | 89.16 | 89.84 | 88.85 | 90.60 | 85.67 | 93.09 | 94.47 | 85.55 | 89.59 | 89.81 |
| Quest (256) | Bud. | 4.99 | 3.66 | 1.07 | 1.98 | 3.54 | 1.63 | 2.48 | 1.82 | 8.90 | 2.16 | 2.03 | 7.96 | 2.36 | 3.43 |
| | Rec. | 57.87 | 56.09 | 34.14 | 47.80 | 58.73 | 42.88 | 47.53 | 39.36 | 74.56 | 57.32 | 63.93 | 67.67 | 45.58 | 53.34 |
| Quest (1k) | Bud. | 20.01 | 14.69 | 4.29 | 7.94 | 14.23 | 6.54 | 9.93 | 7.31 | 34.82 | 8.66 | 8.15 | 31.90 | 9.47 | 13.69 |
| | Rec. | 92.34 | 86.72 | 62.51 | 77.35 | 88.00 | 72.61 | 80.29 | 71.75 | 98.13 | 83.25 | 84.38 | 97.13 | 78.79 | 82.55 |
| Quest (4k) | Bud. | 73.32 | 53.60 | 17.17 | 31.57 | 54.63 | 26.19 | 39.07 | 29.23 | 85.90 | 33.39 | 32.66 | 80.52 | 37.33 | 45.74 |
| | Rec. | 99.30 | 97.14 | 78.23 | 90.21 | 97.67 | 86.98 | 93.46 | 87.95 | 99.79 | 92.95 | 92.73 | 99.44 | 92.01 | 92.91 |
| Quest (8k) | Bud. | 94.75 | 87.44 | 34.33 | 60.65 | 85.83 | 52.32 | 68.77 | 56.33 | 96.15 | 60.88 | 65.36 | 92.59 | 64.54 | 70.76 |
| | Rec. | 99.78 | 99.26 | 88.84 | 96.06 | 99.26 | 93.82 | 97.15 | 95.25 | 99.93 | 97.37 | 97.98 | 99.80 | 96.61 | 97.01 |
| DS (256) | Bud. | 5.02 | 3.68 | 1.07 | 1.99 | 3.57 | 1.64 | 2.49 | 1.83 | 8.95 | 2.17 | 2.04 | 8.00 | 2.37 | 3.45 |
| | Rec. | 46.55 | 46.81 | 31.28 | 40.68 | 47.57 | 36.43 | 36.91 | 34.13 | 55.26 | 45.34 | 51.63 | 49.41 | 33.36 | 42.72 |
| DS (1k) | Bud. | 20.07 | 14.73 | 4.30 | 7.96 | 14.27 | 6.55 | 9.95 | 7.33 | 34.93 | 8.69 | 8.18 | 32.00 | 9.49 | 13.73 |
| | Rec. | 74.63 | 72.47 | 52.89 | 63.82 | 72.92 | 58.63 | 63.08 | 59.97 | 86.19 | 69.72 | 73.93 | 81.77 | 59.41 | 68.42 |
| DS (4k) | Bud. | 73.38 | 53.65 | 17.19 | 31.61 | 54.68 | 26.22 | 39.10 | 29.26 | 85.90 | 33.42 | 32.70 | 80.54 | 37.36 | 45.77 |
| | Rec. | 97.46 | 94.07 | 77.66 | 87.37 | 94.81 | 83.21 | 89.31 | 86.01 | 99.37 | 91.00 | 92.03 | 98.77 | 88.27 | 90.72 |
| DS (8k) | Bud. | 94.76 | 87.47 | 34.35 | 60.69 | 85.85 | 52.36 | 68.80 | 56.35 | 96.14 | 60.91 | 65.40 | 92.60 | 64.56 | 70.79 |
| | Rec. | 99.26 | 97.14 | 88.84 | 96.06 | 99.26 | 93.82 | 97.15 | 95.25 | 99.93 | 97.37 | 97.98 | 99.80 | 96.61 | 96.88 |

*Table 8.* Per-task LongBench scores on LLaMA-3.1-8B-Instruct against additional sparse attention baselines. RaBitQCache achieves the best average score. Method abbreviations: MPIG=MagicPIG, Pyr=PyramidKV, Snap=SnapKV, PQC=PQCache.

| Dataset | RaBitQ | MPIG | Pyr | Snap | PQC | KIVI |
|---|---|---|---|---|---|---|
| 2wikimqa | 41.07 | 46.25 | 39.66 | 32.93 | 45.04 | 43.86 |
| gov_report | 34.96 | 33.56 | 22.78 | 26.81 | 33.94 | 34.93 |
| hotpotqa | 56.16 | 54.28 | 52.87 | 49.32 | 55.52 | 54.37 |
| lcc | 64.74 | 60.91 | 56.69 | 61.98 | 62.70 | 61.93 |
| multi_news | 27.24 | 26.56 | 22.18 | 26.13 | 26.06 | 27.21 |
| multifieldqa_en | 52.97 | 54.94 | 48.75 | 42.14 | 53.27 | 54.47 |
| musique | 27.87 | 27.16 | 23.90 | 21.88 | 30.02 | 30.92 |
| narrativeqa | 29.65 | 29.01 | 28.88 | 20.61 | 30.16 | 29.63 |
| passage_retrieval_en | 99.50 | 99.50 | 99.50 | 98.50 | 99.50 | 99.00 |
| qasper | 46.29 | 45.04 | 29.87 | 31.41 | 43.92 | 43.56 |
| qmsum | 25.25 | 25.33 | 22.84 | 22.27 | 24.67 | 25.17 |
| repobench | 60.83 | 55.16 | 49.49 | 59.71 | 57.60 | 54.91 |
| triviaqa | 91.67 | 91.66 | 88.71 | 90.11 | 92.08 | 91.76 |
| **Avg.** | **50.63** | 49.95 | 45.09 | 44.91 | 50.34 | 50.13 |

*Table 9.* Per-task LongBench results on LLaMA-3.1-70B-Instruct. "Bud./Score" denotes the realized average token budget and the corresponding per-task attention recall for each sparse method.

| Dataset | Score | | | | Budget / Attn. Recall | | |
|---|---|---|---|---|---|---|---|
| | Full | RaBitQ | Quest | SparQ | Quest | RaBitQ | SparQ |
| narrativeqa | 35.03 | 34.66 | 32.26 | 34.30 | 1020.7/0.380 | 3550.5/0.855 | 5959.0/0.864 |
| qasper | 50.96 | 49.89 | 48.09 | 50.15 | 1020.2/0.632 | 1308.5/0.883 | 1275.6/0.829 |
| multifieldqa_en | 54.45 | 54.97 | 51.16 | 54.37 | 1020.5/0.590 | 1595.9/0.873 | 1738.4/0.837 |
| hotpotqa | 64.91 | 64.94 | 61.43 | 63.56 | 1020.6/0.497 | 2289.4/0.863 | 3215.3/0.837 |
| 2wikimqa | 68.17 | 67.70 | 55.77 | 65.46 | 1020.0/0.606 | 1557.5/0.874 | 1793.7/0.841 |
| musique | 44.61 | 44.16 | 40.55 | 37.88 | 1020.6/0.446 | 2690.8/0.850 | 3906.0/0.827 |
| gov_report | 35.03 | 35.21 | 33.67 | 35.19 | 1021.0/0.521 | 2008.0/0.879 | 2569.1/0.849 |
| qmsum | 24.45 | 24.59 | 22.85 | 24.72 | 1021.2/0.426 | 2700.8/0.864 | 3490.1/0.851 |
| multi_news | 26.74 | 26.74 | 26.90 | 27.00 | 994.1/0.782 | 856.9/0.904 | 707.9/0.841 |
| triviaqa | 94.04 | 94.04 | 90.44 | 94.29 | 1020.7/0.587 | 2188.4/0.889 | 2946.3/0.865 |
| passage_retrieval_en | 98.50 | 99.50 | 99.50 | 99.50 | 1020.6/0.642 | 2028.4/0.881 | 3131.9/0.885 |
| lcc | 57.71 | 57.30 | 47.99 | 57.85 | 1020.5/0.695 | 922.8/0.896 | 801.8/0.825 |
| repobench | 55.46 | 55.84 | 48.32 | 54.61 | 1021.3/0.438 | 2196.8/0.871 | 2700.4/0.837 |
| **Avg.** | **54.62** | **54.58** | 50.69 | 53.76 | – | – | – |

*Table 10.* RULER accuracy on LLaMA-3.1-8B-Instruct. DS and Quest use 4096-token fixed budgets; SparQ uses its default; RaBitQCache uses adaptive Top-$p$ with $p = 0.9$ and reports the realized average token budget (last column).

| Length | Full | RaBitQ | DS_4096 | Quest_4096 | SparQ | RaBitQ Budget |
|---|---|---|---|---|---|---|
| 8K | 85.0 | 84.7 | 84.6 | 84.7 | 84.3 | 1076 |
| 16K | 78.6 | **80.3** | 79.2 | 78.6 | 79.0 | 1793 |
| 32K | 76.8 | 76.7 | 77.2 | 75.2 | 76.6 | 2645 |
| 64K | 78.2 | 76.5 | 74.0 | 74.3 | 76.4 | 3581 |
| **Avg.** | **79.65** | 79.55 | 78.75 | 78.20 | 79.08 | 2274 |

*Table 11.* GSM8K accuracy and average attention recall on LLaMA-3.1-8B-Instruct. Recall is normalized by Full Attention.

| Metric | Full | RaBitQ | DS | Quest | SparQ |
|---|---|---|---|---|---|
| Accuracy | 0.81 | 0.77 | 0.77 | 0.70 | 0.77 |
| Recall | 1.00 | **0.888** | 0.727 | 0.813 | 0.605 |

*Table 12.* Per-task LongBench scores with and without centroid re-centering (LLaMA-3.1-8B-Instruct).

| Dataset | RaBitQCache | w/o Re-centering |
|---|---|---|
| 2wikimqa | 41.07 | 42.17 |
| gov_report | 34.96 | 33.94 |
| hotpotqa | 56.16 | 54.44 |
| lcc | 64.74 | 62.84 |
| multi_news | 27.24 | 27.34 |
| multifieldqa_en | 52.97 | 52.48 |
| musique | 27.87 | 29.67 |
| narrativeqa | 29.65 | 29.11 |
| passage_retrieval_en | 99.50 | 99.50 |
| qasper | 46.29 | 44.06 |
| qmsum | 25.25 | 24.74 |
| repobench | 60.83 | 61.22 |
| triviaqa | 91.67 | 91.79 |
| **Avg.** | **50.63** | 50.25 |

