# OpenReview forum: "RaBitQCache: Rotated Binary Quantization for KVCache in Long Context LLM Inference"
_ICML.cc/2026/Conference — ICML 2026 regular_

### Official Review · Reviewer_NSh7 · 2026-02-13

**Soundness:** 2
**Presentation:** 2
**Significance:** 2
**Originality:** 2
**Overall Recommendation:** 4
**Confidence:** 4

**Summary:**

This paper introduces RaBitQCache, a novel sparse attention framework designed to accelerate long-context Large Language Model inference. Authors argue that existing dynamic sparse attention methods often suffer from inaccurate proxy scores and a lack of theoretical guarantees. To address this, RaBitQCache employs randomized rotated binary quantization to compress Key vectors and uses high-throughput INT4-Binary arithmetic to compute an unbiased estimator for attention scores with a proven error bound. This theoretically grounded proxy score enables adaptive Top-p retrieval, which dynamically selects tokens based on the actual attention distribution rather than a fixed budget. Extensive evaluations on the LongBench benchmark demonstrate that RaBitQCache achieves up to 2.16x end-to-end speedup and 3.88x decoding latency reduction compared to full attention and outperforms state-of-the-art baselines like Quest, DS, and SparQ.

**Compliance With Llm Reviewing Policy:**

Affirmed.

**Final Justification:**

The author addressed most of my concern, I recommend it for acceptance.

**Key Questions For Authors:**

1. See weaknesses.
2. Why your method employs an adaptive Top-p strategy (a ratio p), while baseline methods like Quest and DS are evaluated with fixed token budgets (e.g., 256, 1024). Is such comparison fair since retaining more KV cache do not necessarily mean better performance?

**Limitations:**

yes

**Strengths And Weaknesses:**

### Strengths:
1. The paper's presentation is good, with clear structure and well-defined notations, making the complex methodology easy to understand.
2. Moving from a static Top-k to an adaptive Top-p retrieval is effective. It allows the method to dynamically adjust the computational budget based on the inherent sparsity of the attention distribution across different layers and tasks.
3. The experimental evaluation and results are good. Paper has convincing comparisons against baselines and rigorous analysis demonstrating matched accuracy and significant speedups.
4. The efficiency breakdowns and ablation studies are thorough. Authors provide detailed analyses of component contributions, system overhead, and the impact of key hyperparameters.

### Weaknesses:
1. The Johnson-Lindenstrauss Lemma-based estimation is not strongly differentiated from related work in sparse attention. The paper uses a lot of space to deriving this known theoretical foundation without a clear comparison of how RaBitQCache differs from or improves upon related works.
2. The evaluation is limited to the LongBench benchmark. It lacks validation on synthetic benchmarks (e.g., needle-in-a-haystack tests or Ruler). This is crucial for rigorously testing long-context information retention of RaBitQCache.
3. The method's performance is primarily tested on tasks with relatively short-generation answers (e.g., QA, summarization). Its performance in scenarios which require very long output generation (e.g., reasoning in mathematical problems) remains unverified.

---

> ### Author Rebuttal · Authors · 2026-03-31
>
> We sincerely thank the reviewer for the thoughtful feedback, and we will revise the paper accordingly in response to the reviewer’s comments.
>
> **Weakness 1.**
> We thank the reviewer for the careful reading and constructive feedback. The JL lemma itself is not our claimed novelty; rather, it is the theoretical ingredient that enables our unbiased estimator and adaptive Top-p retrieval, and is directly related to whether Theorem A.1 holds. We also want to restate the key difference between our method and prior sparse attention methods. Existing dynamic sparse attention methods rely on proxy scores that can only estimate the relative order of tokens. Consequently, they are restricted to fixed-budget Top-k retrieval. One motivation of our paper is that different tasks require different token budgets, which leads to the need for Top-p sparse attention. To make Top-p feasible, we must estimate the actual magnitude of attention scores, rather than only their relative order. This is why we need a scheme that can efficiently approximate the actual attention score.
>
> **Weakness 2, 3.**
> We appreciate the reviewer’s concern that the original evaluation was limited. To address this, we add experiments on RULER and GSM8K with LLaMA 3.1 8B Instruct. We also test AIME25 on LLaMA 3.1 70B Instruct, but on this dataset almost all methods, including Full attention, achieve nearly zero accuracy, so we do not think that result is informative. In RULER, DS and Quest use a 4096-token budget, SparQ uses its default setting, and RaBitQCache uses adaptive retrieval with p=0.9, for which the resulting token budgets are shown below. In GSM8K, since Full uses an average budget of 776.5, DS and Quest use a 256-token budget. The results show that on RULER our method achieves better performance with fewer tokens, and even surpasses Full in some cases. Notably, it does so dynamically, using an average of 1076 tokens at 8K and scaling naturally to 3581 tokens at 64K. On GSM8K, our method still maintains high recall without sacrificing performance when the output becomes longer.
>
> We also want to discuss the possible centroid shift under long outputs. Theoretically, if the normalized vectors k and q remain approximately uniformly distributed on the unit hypersphere without re-centering, the error can still be bounded by $O(1/\sqrt{D})$; otherwise, the error grows as the deviation increases. In our LongBench ablation, removing centroid re-centering changes the average score only from 50.63 to 50.25, suggesting that the degradation is small. Since this re-centering step is required by the mathematical derivation, while the performance drop is small in our current experiments, whether it is truly necessary in practical applications still needs further exploration. We shall leave this issue for future work.
>
> |RULER|Full|RaBitQCache|DS_4096|Quest_4096|SparQ|
> |---|---|---|---|---|---|
> |8K|85|84.7|84.6|84.7|84.3|
> |16K|78.6|80.3|79.2|78.6|79|
> |32K|76.8|76.7|77.2|75.2|76.6|
> |64K|78.2|76.5|74|74.3|76.4|
>
> |RaBitQCache RULER_Budget||
> |---|---|
> |8K|1076|
> |16K|1793|
> |32K|2645|
> |64K|3581|
>
> |GSM8K|Full|RaBitQCache|DS|Quest|SparQ|
> |---|---|---|---|---|---|
> |accuracy|0.81|0.77|0.77|0.7|0.77|
> |Recall Score|1|0.888|0.727|0.813|0.605|
>
> |Dataset|RaBitQCache|RaBitQCache_No-Recenter|
> |---|---|---|
> |2wikimqa|41.07|42.17|
> |gov_report|34.96|33.94|
> |hotpotqa|56.16|54.44|
> |lcc|64.74|62.84|
> |multi_news|27.24|27.34|
> |multifieldqa_en|52.97|52.48|
> |musique|27.87|29.67|
> |narrativeqa|29.65|29.11|
> |passage_retrieval_en|99.5|99.5|
> |qasper|46.29|44.06|
> |qmsum|25.25|24.74|
> |repobench|60.83|61.22|
> |triviaqa|91.67|91.79|
> |Avg.|50.63|50.25|
>
> **Key Question 2.**
> We appreciate the reviewer’s question regarding the fairness of comparing our adaptive Top-p strategy with baselines evaluated under fixed token budgets. The key point is that Top-p requires estimating the actual attention score to recover a target ratio p, whereas existing methods only estimate relative order and their absolute values are not meaningful. Therefore, these baselines can only be evaluated in their natural fixed-budget setting, while our method provides an unbiased estimate of attention scores with a provable error bound, and therefore can naturally support adaptive Top-p sparse attention. To make the comparison transparent, we also explicitly report the actual budget used by our method. As shown in Table 5, on LongBench RaBitQCache captures 89.85% of the attention score using only 17.33% budget, which is close to the Oracle upper bound of 89.30% with 11.42% budget. In contrast, baseline methods often require larger fixed budgets yet achieve lower accuracy and recall. This highlights the main advantage of our method: the goal is not to keep more KV cache, but to recover as much attention-score mass as possible with the minimum necessary budget. So we think this comparison is not only fair, but also highlights the key advantage of our method.

---

> > ### Author Rebuttal · Reviewer_NSh7 · 2026-04-01
> >
> > The author's response, especially the answers regarding the Ruler dataset and the GSM8K dataset, has resolved my confusion. I have decided to increase the Overall Recommendation to 4.
> >
> > Additionally, the author discussed the role of re-centering and found that its impact is minimal. We would like to point out that recent work such as [1] indicates that K and Q vectors are usually tightly clustered, which means the author's assumption of being uniformly distributed on the unit hypersphere will be invalid.
> >
> > In summary, I believe this work demonstrates certain value and therefore recommend it for acceptance.
> >
> > [1] BEYOND SPEEDUP-UTILIZING KV CACHE FOR SAMPLING AND REASONING. ICLR 2026.

---

> > > ### Author Response · Authors · 2026-04-01
> > >
> > > Dear Reviewer NSh7,
> > >
> > > We are very pleased that our response has resolved your concerns, and we sincerely thank you for your encouraging feedback and for your decision to increase the Overall Recommendation. Your thoughtful comments and suggestions have been very helpful in improving the quality and clarity of our paper.
> > >
> > > We also sincerely thank you for pointing out this recent related work. As discussed in our rebuttal, whether the re-centering operation is truly necessary in practice still requires further study, and we will include this discussion and the newly added experimental results in the final version of the paper. We greatly appreciate the time and effort you have devoted to reviewing our work.
> > >
> > > Finally, we would like to express our sincere gratitude for your valuable feedback, helpful suggestions, and encouragement.
> > >
> > > Best regards,
> > >
> > > RaBitQCache authors

---

### Official Review · Reviewer_rw9o · 2026-03-03

**Soundness:** 2
**Presentation:** 3
**Significance:** 3
**Originality:** 3
**Overall Recommendation:** 5
**Confidence:** 3

**Summary:**

This paper proposes RaBitQCache, a sparse attention framework aimed at fixing the KV cache bottleneck in long-context LLM inference. The main idea is to compress Key vectors into 1-bit representations using randomized rotated binary quantization, while using INT4 for Query vectors. A key contribution is that they derive a proxy score acting as an unbiased estimator with a proven error bound. This allows them to use adaptive Top-p retrieval, dynamically adjusting the token budget based on sparsity instead of sticking to a fixed Top-k. They also implemented some system-level optimizations like asynchronous pipelining for prefill and lazy updates for decoding. Experiments on Llama-3.1-8B and Longchat-7B-v1.5 show it achieves decent speedups (up to 2.16x end-to-end) and saves memory I/O without hurting generation quality.

**Compliance With Llm Reviewing Policy:**

Affirmed.

**Final Justification:**

The rebuttal fully addressed my concerns and I've raised my score. I recommend to accept.

**Key Questions For Authors:**

I have some key questions as follows and I will update the overall recommendation score if some key concerns are addressed.
1. Comparison with other LSH. How does this specific rotated binary quantization compare to standard LSH (SimHash) or other vector quantization methods (PQ, SQ) in terms of recall/bitrate trade-off?
2. Missing Baselines. Why were strong baselines like MagicPIG, PQCache, SnapKV, and KIVI omitted from the comparison? PQCache is particularly relevant as it also targets quantized retrieval. How does RaBitQCache compare to these methods in terms of the accuracy/speed trade-off?

**Limitations:**

yes

**Strengths And Weaknesses:**

# Strengths

## Soundness
The paper offers a rigorous theoretical foundation by deriving an unbiased estimator with proven error bounds. The evaluation is comprehensive, covering 13 LongBench tasks and comparing against baselines like Quest, Double Sparse, and SparQ. The inclusion of an "Oracle" baseline effectively contextualizes the method's recall capabilities.

## Presentation
The manuscript is clearly written and logically structured. The separation between algorithmic contributions (estimator) and system optimizations (asynchronous prefill, lazy updates) is effective. The rationale for adaptive Top-p retrieval is well-articulated.

## Significance
Efficient long-context inference is crucial. RaBitQCache demonstrates meaningful speedups (up to 2.16x) and memory I/O reduction on H20 GPUs while preserving generation quality. The use of 1-bit quantization for index retrieval is a practical and impactful strategy for memory-constrained environments.

## Originality
Applying randomized rotated binary quantization to KV cache compression is novel in this context. The construction of an unbiased estimator specifically for adaptive Top-p retrieval distinguishes this work from standard heuristic approaches.

# Weaknesses

## Soundness
- **Missing Baselines**: Crucially, the evaluation omits key state-of-the-art baselines like MagicPIG, PQCache, SnapK, PyramidK, and KIVI. Given the proliferation of sparse attention methods that often share similar underlying principles (e.g., quantization, LSH, or eviction), a comprehensive comparison against these strong baselines is essential to objectively validate the superiority of the proposed 1-bit rotated quantization over other quantization or eviction strategies.

## Originality
The core technique resembles Locality Sensitive Hashing (LSH). The novelty lies primarily in its specific application to attention and the estimator formulation. System optimizations like "lazy updates" are standard in efficient attention systems (e.g., StreamingLLM), reducing the novelty of the system design aspect.

---

> ### Author Rebuttal · Authors · 2026-03-31
>
> We sincerely thank the reviewer for the insightful feedback, and we will revise the paper accordingly in response to the reviewer’s comments.
>
> **Key Question 1.**
> We would first like to clarify that our method is fundamentally different from locality-sensitive hashing methods. LSH-style methods group similar vectors together so that retrieval is based on relative similarity to the query vector. Since they only preserve relative order, their recall must be evaluated under a fixed budget. In contrast, one main motivation of our paper is that different tasks require different token budgets, which leads to the need for Top-p sparse attention. To make Top-p feasible, we must estimate the actual magnitude of attention scores rather than only their relative order. Our estimator is theoretically grounded as an unbiased estimator of attention scores with a provable error bound. This is why our method cannot be simply replaced by LSH, PQ, or similar alternatives.
>
> To address the concern about recall, we also compare our method with PQ, LSH, and SQ on SIFT1M, using 10,000 queries and recall@100. We use the following settings: LSH with 256 bits, PQ with M = 16 and nbits = 8, and SQ with 4-bit scalar quantization. The raw ANNS recall is shown below. However, we would like to emphasize that this metric is not especially meaningful for our setting. First, in the ANNS literature, quantization methods are typically followed by re-ranking. If re-ranking is allowed, our recall can reach 99.9883%. In LLM inference, however, such re-ranking is not acceptable, because our goal is precisely to reduce attention-score computation rather than to recover exact nearest neighbors. Second, datasets such as SIFT1M contain vectors with much more balanced significance, which makes perfect separation by quantization inherently difficult. In LLM inference, by contrast, attention is highly skewed: a small number of tokens carry most of the attention mass. Therefore, what matters is to recover those important tokens, not to maximize ANNS recall itself. This is exactly what Table 5 in our paper reflects: with only 17.33% budget, our method already captures 89.85% of the attention score, which is close to the Oracle upper bound of 89.30% with 11.42% budget.
>
> |Method|Recall on SIFT1M|
> |---|---|
> |RaBitQCache|62.5707%|
> |PQ|62.7875%|
> |LSH|30.5746%|
> |SQ|86.6317%|
>
>
> **Weakness: Missing Baselines / Key Question 2.**
> We thank the reviewer for raising this important comparison. We add these baselines on LongBench using their official default settings, including MagicPIG (K=10, L=150, W=64), PyramidKV (capacity=512, window=8), SnapKV (capacity=1024, window=64), PQCache (compress=0.1, subvec=2, bits=6), and KIVI (k/v_bits=2, group=32, residual=128). As shown below, RaBitQCache still achieves the best average performance. More importantly, MagicPIG, PyramidKV, SnapKV, and PQCache only provide relative ranking information and therefore can only support fixed-budget Top-k sparse attention. PQCache is similar in this sense: although it uses PQ retrieval, its scores are still derived from k-means centroids and only reflect relative similarity. KIVI, in contrast, is orthogonal to our method: it compresses the KV cache rather than performing query-dependent retrieval, so a natural combination is to first use our method for adaptive selection and then apply KIVI-style compression to the retained KV cache. We shall discuss such an integration as a possible future direction in the paper. Since our method provides an unbiased estimate of attention score, it supports Top-p sparse attention with task-adaptive token budgets, which is the most essential difference from these baselines.
>
> ||RaBitQCache|MagicPIG|PyramidKV|SnapKV|PQCache|KIVI|
> |---|---|---|---|---|---|---|
> |2wikimqa|41.07|46.25|39.66|32.93|45.04|43.86|
> |gov_report|34.96|33.56|22.78|26.81|33.94|34.93|
> |hotpotqa|56.16|54.28|52.87|49.32|55.52|54.37|
> |lcc|64.74|60.91|56.69|61.98|62.70|61.93|
> |multi_news|27.24|26.56|22.18|26.13|26.06|27.21|
> |multifieldqa_en|52.97|54.94|48.75|42.14|53.27|54.47|
> |musique|27.87|27.16|23.90|21.88|30.02|30.92|
> |narrativeqa|29.65|29.01|28.88|20.61|30.16|29.63|
> |passage_retrieval_en|99.50|99.50|99.50|98.50|99.50|99.00|
> |qasper|46.29|45.04|29.87|31.41|43.92|43.56|
> |qmsum|25.25|25.33|22.84|22.27|24.67|25.17|
> |repobench|60.83|55.16|49.49|59.71|57.60|54.91|
> |triviaqa|91.67|91.66|88.71|90.11|92.08|91.76|
> |Avg.|50.63|49.95|45.09|44.91|50.34|50.13|
>
> **Weakness: Originality**
> Our method aims to enable Top-p sparse attention. System techniques such as lazy updates have indeed been used in prior systems, but we do not simply reuse them without modification. These components, such as lazy updates, CUDA kernel implementation, and efficient Top-p retrieval, are necessary system-level optimizations for making our method practical. Although the system novelty of each individual component may be limited, we still view them as necessary parts of the end-to-end realization of our method.

---

> > ### Author Rebuttal · Reviewer_rw9o · 2026-03-31
> >
> > Thanks for your detailed response and solid evaluation results. My concerns are resolved and I will raise my score.

---

> > > ### Author Response · Authors · 2026-04-01
> > >
> > > Dear Reviewer rw9o,
> > >
> > > We are very pleased that our response has resolved your concerns, and we sincerely thank you for your encouraging feedback and for your willingness to raise your score. Your thoughtful comments and suggestions have been very helpful in improving the quality and clarity of our paper.
> > >
> > > Following your suggestions, we will carefully revise the paper and incorporate the additional experimental results and related discussion into the final version. We greatly appreciate the time and effort you have devoted to reviewing our work.
> > >
> > > Finally, we would like to express our sincere gratitude for your valuable feedback and encouragement.
> > >
> > > Best regards,
> > >
> > > RaBitQCache authors

---

### Official Review · Reviewer_hbK5 · 2026-03-09

**Soundness:** 3
**Presentation:** 3
**Significance:** 2
**Originality:** 3
**Overall Recommendation:** 4
**Confidence:** 3

**Summary:**

Long-context large language model inference is hindered by the massive Key-Value (KV) cache, and existing sparse attention methods often rely on static budgets or biased proxy scores. To address this, RaBitQCache introduces a framework using randomized rotated binary quantization and binary-INT4 arithmetic to efficiently estimate attention weights with an unbiased proxy score and provable error bounds, enabling adaptive Top-p retrieval. The system further incorporates hardware-aware optimizations like asynchronous pipelining and lazy updates to mask overhead, resulting in significant acceleration and reduced memory I/O while preserving generation quality.

**Compliance With Llm Reviewing Policy:**

Affirmed.

**Final Justification:**

My final rating is 4. My concern is solved.

**Key Questions For Authors:**

1. The method relies on prefill-phase centroids as proxies for global centroids during quantization. Could the authors provide an ablation that evaluates the impact of removing this centroid-based re-centering step on retrieval recall and generation quality? It would also be informative to assess how this approximation degrades in scenarios involving extended generation, where the decoding distribution may shift significantly from the prefill distribution.
2. The binary quantization component of RaBitQCache is conceptually modular. Have the authors considered replacing it with alternative KV cache quantization schemes, such as rotation-aware methods (e.g., RotateKV [1]) or scalar/asymmetric quantization approaches from the ANNS literature (e.g., SQ)? A comparison along this axis would help clarify whether the performance gains are attributable to the specific properties of RaBitQ-style binary quantization or are more broadly achievable with other quantization backends.

[1] Zunhai Su, Hanyu Wei, Zhe Chen, Wang Shen, Linge Li, Huangqi Yu, and Kehong Yuan. 2025. RotateKV: accurate and robust 2-bit KV cache quantization for LLMs via outlier-aware adaptive rotations (IJCAI '25).

**Limitations:**

See weaknesses.

**Strengths And Weaknesses:**

**Strengths:**

1. The proposed proxy score is theoretically grounded, offering a proven unbiased estimator with a rigorous error bound, which is a meaningful advancement over existing heuristic-based approaches.
2. The asynchronous prefill pipelining strategy is well-motivated and elegantly designed, effectively amortizing the index construction overhead by overlapping it with the compute-intensive dense attention pass, resulting in negligible prefill latency impact.

**Weaknesses:**

1. The random orthogonal rotation introduces non-trivial computational overhead during the prefill phase, particularly the rotation cost. While the authors claim this is masked by asynchronous pipelining, no dedicated analysis is provided to quantify its standalone cost or justify that this overhead remains acceptable under varying hardware and batch size configurations.
2. The ablation studies are notably incomplete, leaving several design choices insufficiently validated. Specifically, the sensitivity of the lazy update buffer size on both efficiency and generation quality is not analyzed. Furthermore, the contribution of the randomized rotation is never isolated. Without these ablations, it is difficult to attribute the observed performance gains to specific components of the proposed framework.

---

> ### Author Rebuttal · Authors · 2026-03-31
>
> We sincerely thank the reviewer for the valuable feedback, and we will revise the paper accordingly in response to the reviewer’s comments.
>
> **Weakness 1.**
> We thank the reviewer for raising the concern about the overhead of random rotation in the prefill phase. In Section 5.3, we report a dedicated prefill performance analysis showing that this overhead remains acceptable in practice. To further address the reviewer’s concern, we additionally include an ablation with and without the asynchronous pipeline. The results show that enabling the pipeline brings up to 1.49× speedup over the non-pipelined variant, confirming that the asynchronous design is effective.
>
> |prefill (ms)|Full|RaBitQCache|RaBitQCache_no_pipeline|
> |---|---|---|---|
> |10k|39.1|40.5|54.6|
> |20k|89.2|94.7|129.3|
> |30k|165.3|183.8|273.0|
>
> **Weakness 2.**
> We agree that the original ablation study was incomplete, and we have added several analyses. Beyond the pipeline experiment above, we study the effect of the lazy-update buffer size on generation quality. We observe that the most recent 5 tokens account for 21.3% of the attention score. Because our retrieval is based on Top-p, when the lazy buffer size is 5, the number of selected tokens can collapse to only 11, leading to a severe performance drop. We also find that the optimal window size varies across tasks. Nevertheless, when the buffer size is in the range of 64–256, the LongBench score fluctuates by less than 1%, indicating a stable region. Second, the rotation matrix is used to construct a random codebook, which is a key foundation of our unbiased estimation. We further replace this random codebook with k-means centroids from key and query vectors. Although one may expect a query-related codebook to work better, the final mean relative error actually increases from 3.5% to 6.3%, further confirming that randomized rotation is necessary. We discuss the remaining ablation in our response to the Key Questions below.
>
> **Key Question 1.**
> The centroid-based re-centering step is introduced for theoretical reasons. If the normalized vectors k and q remain approximately uniformly distributed on the unit hypersphere without re-centering, the error can still be bounded by $O(1/\sqrt{D})$; otherwise, the error grows as the deviation increases. In our ablation, removing re-centering changes the average LongBench score from 50.63 to 50.25. We also add GSM8K for the short-input/long-output setting, where our method still achieves high recall and competitive performance. Since this re-centering step is required by the mathematical derivation, while the performance drop is small in our current experiments, whether it is truly necessary in practical applications still needs further exploration. We shall leave this issue for future work.
>
> | |RaBitQCache|RaBitQCache_No-Recenter|
> |---|---|---|
> |2wikimqa|41.07|42.17|
> |gov_report|34.96|33.94|
> |hotpotqa|56.16|54.44|
> |lcc|64.74|62.84|
> |multi_news|27.24|27.34|
> |multifieldqa_en|52.97|52.48|
> |musique|27.87|29.67|
> |narrativeqa|29.65|29.11|
> |passage_retrieval_en|99.5|99.5|
> |qasper|46.29|44.06|
> |qmsum|25.25|24.74|
> |repobench|60.83|61.22|
> |triviaqa|91.67|91.79|
> |Avg.|50.63|50.25|
>
> |GSM8K|Full|RaBitQCache|DS|Quest|SparQ|
> |---|---|---|---|---|---|
> |accuracy|0.81|0.77|0.77|0.70|0.77|
> |Recall Score|1|0.888|0.727|0.813|0.605|
>
> **Key Question 2.**
> Our paper is motivated by introducing a practical Top-p sparse attention mechanism. This requires the proxy score to estimate actual attention scores, rather than merely preserve relative order. In this sense, PQ- or SQ-style quantization backends may be applicable to fixed-budget Top-k retrieval, but they only provide relative score information and therefore cannot support Top-p selection in our setting. RotateKV addresses a different aspect of the problem: it mainly focuses on robust low-bit KV-cache quantization, whereas our method targets query-dependent token selection for adaptive Top-p sparse attention. As a result, the two are largely complementary rather than direct substitutes. A natural combination is to first use our method to identify the important tokens, and then apply a RotateKV-style quantization scheme to the corresponding KV cache, thereby potentially obtaining both retrieval-side acceleration and storage-side reduction. We shall clarify this point in the paper and discuss such an integration as a possible future direction. For fixed-budget sparse attention, PQCache is the most relevant baseline. We evaluate PQCache on LongBench and find that RaBitQCache achieves better average performance while also supporting adaptive Top-p sparse attention.
>
> | |RaBitQCache|PQCache|
> |---|---|---|
> |2wikimqa|41.07|45.04|
> |gov_report|34.96|33.94|
> |hotpotqa|56.16|55.52|
> |lcc|64.74|62.7|
> |multi_news|27.24|26.06|
> |multifieldqa_en|52.97|53.27|
> |musique|27.87|30.02|
> |narrativeqa|29.65|30.16|
> |passage_retrieval_en|99.5|99.5|
> |qasper|46.29|43.92|
> |qmsum|25.25|24.67|
> |repobench|60.83|57.6|
> |triviaqa|91.67|92.08|
> |Avg.|50.63|50.34|

---

> > ### Author Rebuttal · Reviewer_hbK5 · 2026-04-03
> >
> > I have no more questions. Good luck.

---

> > > ### Author Response · Authors · 2026-04-03
> > >
> > > Dear Reviewer hbK5,
> > >
> > > We are delighted that our response has successfully addressed your concerns, and we deeply appreciate your positive and encouraging feedback. Your constructive comments have been instrumental in enhancing the overall quality of our work.
> > >
> > > As advised, we will update the paper to include the newly provided experimental results and the corresponding in-depth discussions. We are truly grateful for the dedication and time you invested in evaluating our submission.
> > >
> > > Once again, thank you for your continued support and invaluable insights.
> > >
> > > Best regards,
> > >
> > > RaBitQCache authors

---

### Official Review · Reviewer_bPLt · 2026-03-11

**Soundness:** 3
**Presentation:** 3
**Significance:** 3
**Originality:** 2
**Overall Recommendation:** 4
**Confidence:** 4

**Summary:**

This paper studies long-context LLM inference under the KV-cache bottleneck and proposes RabitQCache, a sparse attention framework for retrieving only a subset of tokens during decoding. The method builds a binary index over rotated and normalized key vectors during prefill, then estimates attention relevance at decode time using a lightweight binary-key / INT4-query computation together with a per-token correction factor. Based on this proxy score, the method performs adaptive Top-p retrieval rather than using a fixed token budget, and then computes attention over the retrieved tokens plus a local window. In addition to the algorithmic component, the paper includes a hardware-aware implementation with asynchronous prefill pipelining, lazy decode updates, and optimized kernels for binary-INT4 score computation and Top-p masking.

**Compliance With Llm Reviewing Policy:**

Affirmed.

**Key Questions For Authors:**

Please refer to my weaknesses part.

**Limitations:**

yes

**Strengths And Weaknesses:**

### Strengths

- The paper addresses an important and practical problem. Long-context inference is increasingly limited by KV-cache bandwidth and memory movement rather than by model quality alone, so improving token selection efficiency is a relevant contribution.
- The method is well motivated. The paper clearly argues that fixed-budget sparse retrieval can be suboptimal because attention sparsity varies across heads, layers, and tasks, and it proposes adaptive Top-p retrieval as a more flexible alternative.
- The submission combines algorithmic and systems ideas in a coherent way. The rotated binary indexing scheme, the lightweight proxy score, the correction factor, and the optimized execution path are tied together into a single end-to-end serving design rather than being presented as isolated tricks.

### Weaknesses
- The paper’s key theoretical narrative is that the proxy score is an unbiased estimator that supports adaptive Top-p retrieval. However, the proof appears to rely on randomness over the orthogonal rotation, whereas the implementation samples the rotation matrix once during initialization and then keeps that realized index fixed for serving. In other words, the proof seems to establish an average-over-random-rotations statement, while the system actually uses one fixed realization. That leaves a meaningful gap between the proven object and the deployed object, especially because Top-p retrieval depends on the calibration of a concrete score distribution at inference time rather than on an average statement over hypothetical random draws. This does not mean the method fails empirically, but it weakens the extent to which the current proof fully supports the strongest claims in the paper.

- The empirical evaluation, while promising, is still somewhat limited for the breadth of the claims. The experiments use two models and a single H20-based setup, and the paper argues for broad generalizability across tasks and varying sparsity patterns. I would have found the significance stronger with evidence on a wider range of model families, larger contexts, and possibly more deployment environments. As written, the results are encouraging, but the generality claim feels somewhat ahead of the evidence.

---

> ### Author Rebuttal · Authors · 2026-03-31
>
> We sincerely thank the reviewer for the constructive feedback, and we will revise the paper accordingly in response to the reviewer’s comments.
>
> **Weakness 1.**
> We thank the reviewer for pointing out the gap between the theoretical statement and the deployed system. Our use of the random orthogonal rotation is to instantiate a random codebook: a randomly rotated set of hypercube vertices on the unit sphere. In actual inference, this codebook must remain fixed within one run; otherwise, the proxy score and index would not be consistent across prefill and decode. Fixing the realized codebook is therefore a requirement of the method itself, rather than an implementation choice. Therefore, we can pre-generate a pool of random orthogonal matrices offline and randomly choose one at serving time, which preserves the statistical role of randomness while avoiding per-request generation overhead. To further validate this design, we also replace the random codebook with a data-dependent one obtained by k-means centroids from key and query vectors. Although one may expect a query-related codebook to work better, the mean relative error actually increases from 3.5% to 6.3%. Moreover, Table 5 in our paper shows that RaBitQCache already captures 89.85% of the attention score using only 17.33% budget, which is close to the Oracle upper bound of 89.30% with 11.42% budget. Together, these results suggest that, although our current guarantee is stated over the random rotation, the fixed codebook realization used in deployment still behaves consistently in practice.
>
> **Weakness 2.**
> We agree that the original empirical scope was limited. To strengthen the generality claim, we add experiments on another GPU based on Hopper architecture, including synthetic long-context evaluation on RULER, mathematical reasoning on GSM8K, and larger-model evaluation on LLaMA 3.1 70B Instruct. On RULER, DS and Quest use a fixed budget of 4096 tokens, SparQ uses its default setting, and RaBitQCache uses adaptive retrieval with p=0.9, for which the resulting token budgets are shown below. Our method remains highly competitive while using fewer tokens, and even surpasses Full in some settings. On GSM8K, since the Full method has an average budget of 776.5, DS and Quest use a 256-token budget. RaBitQCache matches the accuracy of DS and SparQ while obtaining the highest recall among sparse baselines. We also evaluate LLaMA 3.1 70B Instruct on LongBench. DS is not included here because its official implementation does not support this model. RaBitQCache achieves an average score of 54.58, almost identical to Full (54.62), and clearly outperforms Quest (50.69) and SparQ (53.76). We further report the corresponding budget/score statistics below. Compared with Quest, RaBitQCache achieves consistently higher scores; compared with SparQ, it often uses a smaller budget with comparable or better scores. More importantly, the budget changes substantially across tasks, e.g., 856.9 on multi_news and 922.8 on lcc, but 3550.5 on narrativeqa and 2700.8 on qmsum. This directly shows the advantage of Top-p sparse attention: the budget adapts to the task instead of being fixed in advance.
>
> |RULER|Full|RaBitQCache|DS_4096|Quest_4096|SparQ|
> |---|---|---|---|---|---|
> |8K|85|84.7|84.6|84.7|84.3|
> |16K|78.6|80.3|79.2|78.6|79|
> |32K|76.8|76.7|77.2|75.2|76.6|
> |64K|78.2|76.5|74|74.3|76.4|
>
> |RaBitQCache RULER_Budget||
> |---|---|
> |8K|1076|
> |16K|1793|
> |32K|2645|
> |64K|3581|
>
> |GSM8K|Full|RaBitQCache|DS|Quest|SparQ|
> |---|---|---|---|---|---|
> |accuracy|0.81|0.77|0.77|0.70|0.77|
> |Recall Score|1|0.888|0.727|0.813|0.605|
>
> |LongBench in LLaMA 3.1 70B Instruct|Full|Quest|RaBitQCache|SparQ|
> |---|---|---|---|---|
> |narrativeqa|35.03|32.26|34.66|34.30|
> |qasper|50.96|48.09|49.89|50.15|
> |multifieldqa_en|54.45|51.16|54.97|54.37|
> |hotpotqa|64.91|61.43|64.94|63.56|
> |2wikimqa|68.17|55.77|67.70|65.46|
> |musique|44.61|40.55|44.16|37.88|
> |gov_report|35.03|33.67|35.21|35.19|
> |qmsum|24.45|22.85|24.59|24.72|
> |multi_news|26.71|26.90|26.74|27.00|
> |triviaqa|94.04|90.44|94.04|94.29|
> |passage_retrieval_en|98.50|99.50|99.50|99.50|
> |lcc|57.71|47.99|57.30|57.85|
> |repobench|55.46|48.32|55.84|54.61|
> |Avg.|54.62|50.69|54.58|53.76|
>
> |LongBench in LLaMA 3.1 70B Instruct|Quest Budget/Attn Score|RaBitQCache Budget/Attn Score|SparQ Budget/Attn Score|
> |---|---|---|---|
> |narrativeqa|1020.7/0.380|3550.5/0.855|5959.0/0.864|
> |qasper|1020.2/0.632|1308.5/0.883|1275.6/0.829|
> |multifieldqa_en|1020.5/0.590|1595.9/0.873|1738.4/0.837|
> |hotpotqa|1020.6/0.497|2289.4/0.863|3215.3/0.837|
> |2wikimqa|1020.0/0.606|1557.5/0.874|1793.7/0.841|
> |musique|1020.6/0.446|2690.8/0.850|3906.0/0.827|
> |gov_report|1021.0/0.521|2008.0/0.879|2569.1/0.849|
> |qmsum|1021.2/0.426|2700.8/0.864|3490.1/0.851|
> |multi_news|994.1/0.782|856.9/0.904|707.9/0.841|
> |triviaqa|1020.7/0.587|2188.4/0.889|2946.3/0.865|
> |passage_retrieval_en|1020.6/0.642|2028.4/0.881|3131.9/0.885|
> |lcc|1020.5/0.695|922.8/0.896|801.8/0.825|
> |repobench|1021.3/0.438|2196.8/0.871|2700.4/0.837|

---

> > ### Author Rebuttal · Reviewer_bPLt · 2026-04-04
> >
> > Thank you for the rebuttal.
> >
> > On weakness 1, The fixed-codebook justification, offline rotation pool proposal, and empirical validation sufficiently address my concern.
> >
> > On weakness 2, The additional experiments on Hopper, RULER, GSM8K, and llama 3.1 70b, along with the task-adaptive budget variation, support the generality claim.
> >
> > Both weaknesses are resolved.

---

> > > ### Author Response · Authors · 2026-04-04
> > >
> > > Dear Reviewer bPLt,
> > >
> > > We are delighted that our response has successfully addressed your concerns, and we deeply appreciate your positive and encouraging feedback. Your constructive comments have been instrumental in enhancing the overall quality of our work.
> > >
> > > As advised, we will update the paper to include the newly provided experimental results and the corresponding in-depth discussions. We are truly grateful for the dedication and time you invested in evaluating our submission.
> > >
> > > Once again, thank you for your continued support and invaluable insights.
> > >
> > > Best regards,
> > >
> > > RaBitQCache authors

---

### Decision · Program_Chairs · 2026-04-30

**Decision:**

Accept (regular)

**Comment:**

This paper proposes RaBitQCache, a sparse attention framework using randomized rotated binary quantization to enable adaptive Top-p retrieval with theoretically grounded unbiased estimation. Initial concerns regarding the gap between average-case theory and fixed-codebook deployment, limited evaluation (missing RULER, GSM8K, and strong baselines), and insufficient ablations were fully resolved in the rebuttal: the authors clarified the fixed codebook is a deliberate design choice, added comprehensive experiments on synthetic and reasoning benchmarks, included comparisons with omitted baselines, and provided ablations validating the pipeline and rotation necessity. All reviewers acknowledged their concerns were addressed with unanimous positive recommendations, leading to the final recommendation of Accept.